# The kinetochore prevents centromere-proximal crossover recombination during meiosis

Nadine Vincenten[1], Lisa-Marie Kuhl[2], Isabel Lam[3], Ashwini Oke[4], Alastair RW Kerr[1], Andreas Hochwagen[5], Jennifer Fung[4], Scott Keeney[3], Gerben Vader[2*†], Adèle L Marston[1*†]

[1]The Wellcome Trust Centre for Cell Biology, Institute of Cell Biology, School of Biological Sciences, The University of Edinburgh, Edinburgh, United Kingdom; [2]Department of Mechanistic Cell Biology, Max Planck Institute of Molecular Physiology, Dortmund, Germany; [3]Howard Hughes Medical Institute, Memorial Sloan Kettering Cancer Center, New York, United States; [4]Department of Obstetrics, Gynecology and Reproductive Sciences, Center of Reproductive Sciences, University of California, San Francisco, San Francisco, United States; [5]Department of Biology, New York University, New York, United States

*For correspondence: gerben. vader@mpi-dortmund.mpg.de (GV); adele.marston@ed.ac.uk (ALM)

†These authors contributed equally to this work

Competing interests: The authors declare that no competing interests exist.

**Abstract** During meiosis, crossover recombination is essential to link homologous chromosomes and drive faithful chromosome segregation. Crossover recombination is non-random across the genome, and centromere-proximal crossovers are associated with an increased risk of aneuploidy, including Trisomy 21 in humans. Here, we identify the conserved Ctf19/CCAN kinetochore sub-complex as a major factor that minimizes potentially deleterious centromere-proximal crossovers in budding yeast. We uncover multi-layered suppression of pericentromeric recombination by the Ctf19 complex, operating across distinct chromosomal distances. The Ctf19 complex prevents meiotic DNA break formation, the initiating event of recombination, proximal to the centromere. The Ctf19 complex independently drives the enrichment of cohesin throughout the broader pericentromere to suppress crossovers, but not DNA breaks. This non-canonical role of the kinetochore in defining a chromosome domain that is refractory to crossovers adds a new layer of functionality by which the kinetochore prevents the incidence of chromosome segregation errors that generate aneuploid gametes.

## Introduction

The formation of haploid reproductive cells during meiosis relies on the accurate segregation of chromosomes during two meiotic divisions (meiosis I and II). Faithful segregation of homologous chromosomes during meiosis I is contingent on inter-homologue linkages that are established during the preceding G2/prophase. These linkages (chiasmata) are the final outcome of programmed DNA break formation and crossover (CO) repair. Improper placement of COs in the vicinity of centromeres negatively influences meiotic chromosome segregation (*Hassold and Hunt, 2001*; *Koehler et al., 1996*; *Rockmill et al., 2006*), so CO formation close to centromeres is infrequent in many species, including humans (*Centola and Carbon, 1994*; *Copenhaver et al., 1999*; *Ellermeier et al., 2010*; *Gore et al., 2009*; *Lambie and Roeder, 1986*; *Mahtani and Willard, 1998*; *Nakaseko et al., 1986*; *Puechberty et al., 1999*; *Saintenac et al., 2009*; *Tanksley et al., 1992*). However, the mechanisms that control DNA break formation and CO repair close to centromeres remain poorly understood (*Choo, 1998*; *Talbert and Henikoff, 2010*).

**eLife digest** The cells of animals, plants and many other organisms store most of their DNA inside the cell nucleus, packaged into structures called chromosomes. Most cells contain two copies of each chromosome – one inherited from each parent. However, sex cells (such as egg cells and sperm cells) contain just one copy of every chromosome, so that when they fuse, the new cell that is formed contains the full set.

Sex cells form in a process called meiosis, where a cell containing two copies of every chromosome duplicates its genetic material and then divides to form four new cells, each of which contains one copy of each chromosome. During meiosis, different versions of the same chromosome are able to swap sections of their DNA in a process called crossover. However, if a crossover occurs in the wrong part of the chromosome, the chromosome copies may not segregate correctly during cell division. This can lead to the formation of sex cells that contain the wrong number of chromosome copies, which can cause developmental conditions such as Down's syndrome.

Crossovers tend not to occur at a region of the chromosomes called the centromere, which is where copies of the same chromosome from the same parent are joined together until it is time for them to separate. If a crossover does occur in this region, segregation problems are more likely to occur. However, exactly how crossovers are suppressed at centromeres is not understood.

Vincenten et al. examined crossover positioning in budding yeast cells, which are often used as a model to investigate processes such as cell division. This revealed that a protein complex called Ctf19 stops DNA breaks from occurring near the centromere, and so prevents crossovers. Ctf19 also promotes the enrichment of another protein complex called cohesin near centromeres. This does not prevent DNA breaks from occurring, but also prevents crossovers.

Identifying this role of the Ctf19 complex has paved the way for understanding exactly how DNA breaks are prevented near centromeres. This will allow researchers to determine the impact of misplaced crossovers on how chromosomes segregate into sex cells.

Centromeres are functionally conserved but structurally diverse, ranging from the simple so-called "point" centromeres of budding yeasts to a variety of more complex "regional" or even holocentric centromeres in other eukaryotes (*Allshire and Karpen, 2008*). Point centromeres are defined by a short ~125bp sequence upon which the kinetochore assembles, while regional centromeres are typically comprised of specialized centromeric chromatin interspersed with blocks of heterochromatin. In fission yeast and *Drosophila*, the integrity of pericentromeric heterochromatin was found to repress double-strand break (DSB) formation and centromere-proximal recombination (*Ellermeier et al., 2010*; *Westphal and Reuter, 2002*). Although budding yeast lack pericentromeric heterochromatin, suppression of centromere-proximal recombination is also observed in this organism (*Lambie and Roeder, 1986*, *Chen et al., 2008*). Moreover, pericentromeric CO suppression has been observed in situations where centromere position is uncoupled from its associated heterochromatin, such as in *Drosophila* strains carrying translocated chromosomes (*Mather, 1939*). These observations suggest the existence of a fundamental mechanism of recombination suppression that functions independently of associated heterochromatin.

Genome-wide DSB maps in budding yeast have inferred that the centromere exerts a zone of inhibition of meiotic DSB formation, the activity of which decreases over a distance of approximately 10 kb (*Blitzblau et al., 2007*; *Buhler et al., 2007*; *Pan et al., 2011*). Excision of a centromere relieved this DSB suppression, indicating that the centromere, or its associated factors, exert this effect (*Robine et al., 2007*). The synaptonemal complex component, Zip1 (*Chen et al., 2008*), and the Bloom's helicase, Sgs1 (*Rockmill et al., 2006*), which influences repair pathway choice, are also known to minimize centromere recombination. However, both proteins affect recombination globally, acting at a step after DSB formation, and are not specifically localized at centromeres. Instead, centromere-bound factors are likely to dictate the region of recombination suppression in the surrounding pericentromere through mechanisms that remain unclear.

Candidate centromere-bound factors for the repression of pericentromeric recombination are components of the kinetochore, a sophisticated multi-subunit protein complex nucleated by

centromeric chromatin (reviewed in *Biggins (2013)*; *Cheeseman, (2014)*). Within kinetochores, multiple generally conserved sub-complexes can be recognized that perform specific roles. Outer kinetochore sub-complexes together form an interface with microtubules and serve as a platform for spindle assembly checkpoint signaling, coupling chromosome-microtubule interactions with cell cycle progression. Inner kinetochore sub-complexes direct assembly of the outer kinetochore. Several kinetochore subcomplexes together assemble into a Constitutive Centromere-Associated Network (CCAN; also known as the Ctf19 complex in budding yeast) (reviewed in *McAinsh and Meraldi (2011)*; *Westermann and Schleiffer (2013)*) As its name implies, the CCAN/Ctf19 complex is bound to centromeric chromatin throughout the mitotic or meiotic cell division program. In meiotic G2/prophase of budding yeast, when recombination occurs, only the Ctf19 and Mis12/MIND (Mtw1 including Nnf1-Nsl1-Dsn1) kinetochore complexes are bound to the centromere (*Meyer et al., 2015*; *Miller et al., 2012*). The Ctf19 complex exerts long range effects by promoting cohesin enrichment throughout the ~20–50 kb surrounding pericentromere despite being restricted to the core ~125 bp centromere sequence (*Eckert et al., 2007*; *Fernius and Marston, 2009*; *Ng et al., 2009*). It does so by targeting the Scc2/4 cohesin loader to the centromere, from where cohesin spreads into the pericentromere (*Fernius et al., 2013*). These characteristics make the Ctf19 complex a particularly good candidate for mediating kinetochore-derived recombination suppression. Here we show that both cohesin-independent suppression of DSB formation and cohesin-dependent repair pathway choice underlie a central role for the Ctf19 complex in suppression of CO formation in the pericentromere.

## Results

### The Ctf19 kinetochore subcomplex suppresses pericentromeric COs

To understand how pericentromeric COs are prevented, we used a fluorescent CO reporter assay (*Thacker et al., 2011*) (*Figure 1A*) to measure recombination rates within a pericentromere (around *CEN8,* the centromere of chromosome *VIII*) or, as a control, on a chromosome arm interval of equivalent size on chromosome *VIII* of budding yeast (*Figure 1B,C*). In wild-type cells, map distance, a measure of CO frequency, was 7.5 cM within the arm interval but only 0.04 cM within the pericentromere interval. In cells lacking the synaptonemal component, Zip1, map distance within the pericentromeric interval rose to ~2 cM (*Figure 1B*), in agreement with previous observations (*Chen et al., 2008*), while we observed a modest decrease in map distance within the chromosomal arm region (*Figure 1C*). Thus, the fluorescent reporter assay can report on pericentromeric CO formation.

Next, we tested whether the kinetochore (*Figure 1D*) affects CO formation at pericentromeres. During meiotic prophase, when recombination occurs, only the MIND and Ctf19 complexes are assembled on centromeres (*Meyer et al., 2015*; *Miller et al., 2012*) and their components associate with kinetochores at least partially independently (*Figure 1—figure supplement 1*). We were unable to test the requirement for the MIND complex in preventing pericentromeric recombination using the fluorescent reporter assay because depletion of two components of the essential MIND complex, Dsn1 or Mtw1, prevented proper execution of the meiotic divisions and tetrad formation (*Figure 1—figure supplement 1*). Therefore, we focused on the conserved Ctf19/CCAN kinetochore complex. Using the live cell reporter assay, we observed a significantly increased frequency of pericentromeric COs in cells lacking the Ctf19 complex components Iml3[CENP-L], Chl4[CENP-N], Mcm21[CENP-O] and Ctf19[CENP-P] (*Figure 1B*, *Supplementary file 1*; the gene names of the human homologues are indicated in superscript). This effect appeared to be specific to the pericentromere, as no significant changes in recombination were observed within the chromosomal arm interval in the absence of Iml3[CENP-L], Chl4[CENP-N], Mcm21[CENP-O] and Ctf19[CENP-P] (*Figure 1C*, *Supplementary file 2*). Other kinetochore subunits (Cnn1[CENP-T], Wip1[CENP-W], Nkp1, Nkp2) had a more modest effect on pericentromeric COs, while we found no evidence that Mhf1[CENP-S] and Mhf2[CENP-X], which have additional roles in meiotic DNA repair together with Mph1 (*Osman and Whitby, 2013*), are required for suppression of pericentromeric COs. Thus, the Ctf19 inner kinetochore subcomplex affects pericentromeric meiotic recombination.

We corroborated these findings by analyzing the effect of the Ctf19 complex on global meiotic recombination patterns, using high throughput sequencing to identify single nucleotide polymorphisms in the haploid progeny generated from meiosis of a hybrid yeast strain (*Oke et al., 2014*) (*Figure 2A*). This method allows the detection of CO and non-crossover (NCO) repair products (i.e.

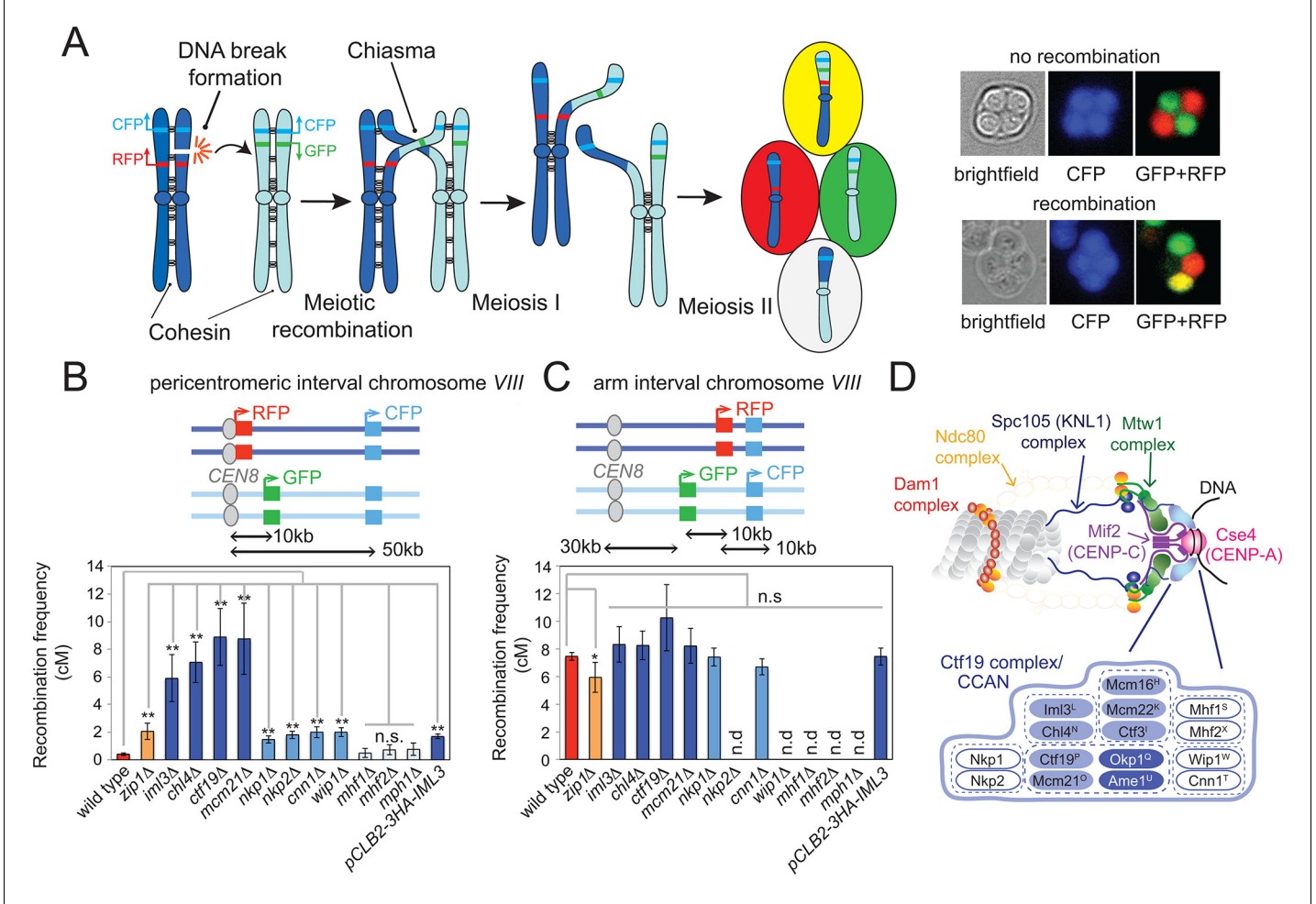

**Figure 1.** The Ctf19 kinetochore sub-complex represses pericentromeric meiotic recombination. (**A**) Scheme of meiosis and the live cell reporter assay to measure CO recombination. Homologous chromosomes are shown in light and dark blue with Green fluorescent protein (GFP), tdTomato (RFP) and m-Cerulean (CFP) reporters represented in green, red and cyan, respectively. Expression of reporters in spores leads to segregation of coloured markers as indicated in the images of live tetrads. (**B, C**) Map distances (centiMorgans (cM)) and standard error (bars) were determined for a ~10 kb pericentromeric (**B**) or chromosomal arm (**C**) interval as described in Materials and methods. p-values were obtained using Fisher's exact test (* $p < 0.05$; ** $p < 0.0001$). (**D**) Schematic representation of the kinetochore showing yeast Ctf19 sub-complex components with superscripts indicating the centromere protein (CENP) equivalent in humans. Proteins essential for vegetative growth or proper spore viability after meiosis are shown in dark and light blue, respectively. CO, crossover; n.d., not determined.

The following figure supplement is available for figure 1:

**Figure supplement 1.** Partial inter-dependence of the Ctf19 complex and MIND subunits.

gene conversions), which can inform on altered regulation of meiotic DNA break repair (*Figure 2B*). Poor spore viability (typically <30%) precluded us from using Ctf19 complex deletion mutants. Instead, we used a meiosis-specific hypomorphic depletion allele of *IML3* (*pCLB2-3HA-IML3*; *Figure 2—figure supplement 1*) and were able to isolate 8 four-spore-viable tetrads for high-throughput sequencing and global recombination analysis (*Supplementary file 3*). Although we observed no global change in the average number of COs or NCOs in the *pCLB2-3HA-IML3* strain compared to wild type (*Figure 2C,D*), the distribution of recombination was affected: the frequency of both COs and NCOs within 20 kb of centromeres was overall significantly increased (*Figure 2E,F*). These events were detected on most, but not all, chromosomes, although we are unable to test significance for individual chromosomes due to an insufficient number of events analyzed (*Figure 2—figure supplement 2A*; *Supplementary file 2*). We note that, in the live cell recombination reporter

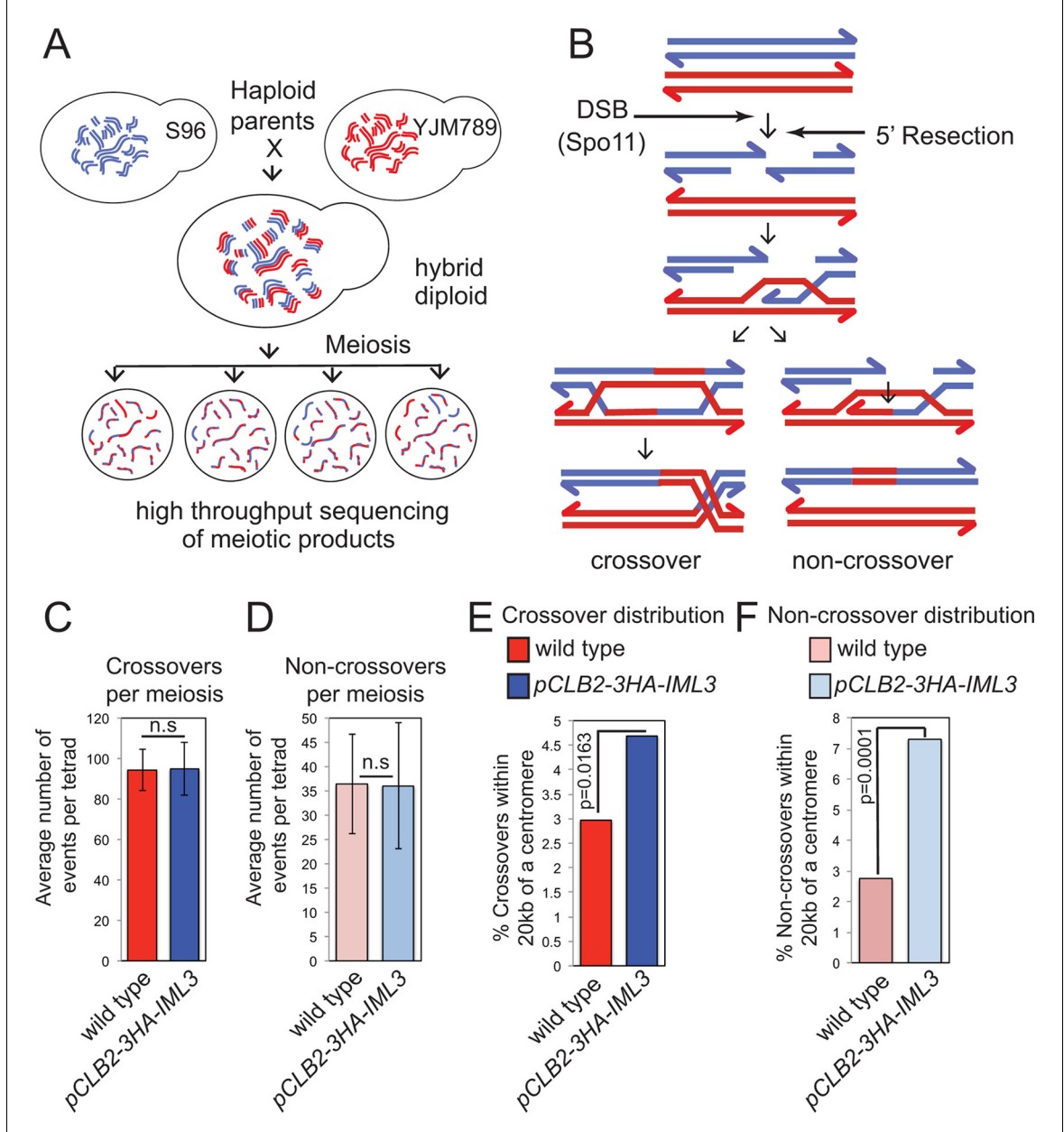

**Figure 2.** Genome-wide analysis shows that a functional Ctf19 complex is required to prevent both pericentromeric COs and NCOs. (**A**) Assay to measure meiotic recombination genome-wide by analysis of SNPs after high-throughput sequencing of germinated spores resulting from hybrid meiosis. (**B**) Scheme of meiotic recombination showing CO and NCO outcomes. (**C, D**) Overall recombination levels are not affected by the *pCLB2-3HA-IML3* mutation. The average numbers of COs (**C**) and NCOs (**D**) per tetrad are not significantly different between wild type and *pCLB2-3HA-IML3* cells. Error bars represent standard deviation. A two-tailed t test indicated non-significance (p>0.05). (**E, F**) Both COs (**E**) and NCOs (**F**) within 20 kb of the pericentromere are increased in *pCLB2-3HA-IML3* cells. Data for wild type is from (*Oke et al., 2014*). Number of meioses scored was 8 for *pCLB2-3HA-IML3* and 52 for wild type. p-values were calculated using chi-square test with Yates correction. CO, crossover; DSB, double strand break; NCO, non-crossover; SNPs, single nucleotide polymorphisms.

The following figure supplements are available for figure 2:

**Figure supplement 1.** Depletion of Iml3 during meiosis.

**Figure supplement 2.** Depletion of Iml3 in meiosis increases the frequency of recombination events within 20 kb of the centromere.

assay, the increase in centromere-proximal COs on chromosome *VIII* was more modest in *pCLB2-3HA-IML3* cells than *iml3△* cells (*Figure 1B,C*), indicating that analysis of the hypomorphic *pCLB2-3HA-IML3* strain likely underestimates the importance of the Ctf19 complex in suppressing pericentromeric recombination events. Nevertheless, the observed effect of *pCLB2-3HA-IML3* on pericentromeric recombination was greater than *sgs1△* (*Figure 2—figure supplement 2B,C*), which is known to affect meiotic recombination near centromeres (*Rockmill et al., 2006*; *Oke et al., 2014*). Together, these experiments demonstrate that the Ctf19 complex shapes the meiotic recombination landscape by minimizing pericentromeric CO recombination.

## The Ctf19 complex inhibits centromere-proximal DNA breaks

We next addressed how the Ctf19 complex prevents pericentromeric COs and NCOs. All meiotic recombination events begin with the programmed introduction of DSBs by the topoisomerase-related protein Spo11 (*Keeney et al., 1997*). DSBs form throughout the genome, but are strongly repressed within ~3 kb of budding yeast centromeres (*Blitzblau et al., 2007*; *Buhler et al., 2007*; *Pan et al., 2011*). We generated genome-wide DSB maps of the *mcm21△* mutant by high-throughput sequencing of oligonucleotides that remain covalently bound to Spo11 as a by-product of DSB processing (*Pan et al., 2011*) (*Figure 3A*). Interestingly, increased centromere-proximal DSBs were detected in the *mcm21△* mutant compared with wild type, revealing a general role for the Ctf19 complex in suppression of pericentromeric DSBs (*Figure 3B,C*). Pericentromeric DSBs in the *mcm21△* mutant reached a level similar to the genome average (dotted lines, *Figure 3C*), indicating that there is no residual DSB suppression in the absence of Mcm21. Strikingly, the increase in DSBs occurred over a narrower domain (~3 kb on each side of the centromere, i.e. a total region of ~6 kb) than CO formation (~20 kb each side of the centromere, ~40 kb in total; *Figure 2E,F*), indicating that suppression of COs at centromeres is not solely due to a reduction in DSB levels. To compare the effect on DSBs and COs more directly, we examined DSB formation in the intervals on chromosome *VIII* analyzed in the live cell recombination assay (*Figure 3D*). As expected, we observed an increase in DSBs within the same pericentromeric interval in which we measured CO frequency using the live cell reporter assay. However, DSBs were increased only ~5-fold over wild type in *mcm21△* within this region (*Figure 3E*), while we observed an ~21-fold increase in COs within the same region (*Figure 3F*; *Figure 1B,C*). Therefore, the increase in DSBs can only account for approximately 24% of the increase in pericentromeric COs.

The DSB effect varied per individual chromosome (*Figure 3—figure supplement 1*). The variability in susceptibility of the different pericentromeres to DSBs is likely explained by their underlying features, since it is well-established that DSB formation is influenced by chromatin and genome organization (e.g. the availability of gene promoters that serve as a preferred target of Spo11 (*Blitzblau et al., 2007*; *Pan et al., 2011*)). Chromosome *I* in particular, showed emergence of a very prominent DSB hotspot in a promoter region immediately adjacent to the centromere (*CEN1*; *Figure 4A*). We exploited the appearance of this hotspot to monitor ectopic DSBs close to centromeres by Southern blotting in the repair-deficient *dmc1△* background, where DSBs persist. Close to *CEN1*, DSBs were detected in the absence of Mcm21[CENP-O] or Ctf19 [CENP-P] (*Figure 4B*). Importantly, DSBs that formed near *CEN1* in the absence of Mcm21[CENP-O] or Ctf19 [CENP-P] were dependent on the catalytic activity of Spo11, demonstrating that these were genuine programmed DSBs (*Figure 4B*). Screening additional Ctf19 complex subunits showed a striking correlation between increased DSB formation at *CEN1* and increased CO formation as measured in our live cell recombination reporter assay (*Figure 4B*, *Figure 4—figure supplement 1A*). Interestingly, depletion of the MIND complex component, Dsn1 (*Figure 1—figure supplement 1*) also resulted in the appearance of *CEN1*-proximal DSBs, suggesting that the overall integrity of the kinetochore might be generally important for repressing DSB formation within pericentromeres. In conclusion, one likely mechanism by which the Ctf19 complex prevents pericentromeric recombination is via the inhibition of DSB formation close to centromeres.

## Pericentromeric cohesin shields against CO repair of centromere-proximal DSBs

During mitotic growth, the Ctf19 complex targets loading of the sister-chromatid-linking complex, cohesin, to the centromere prior to S phase to enrich cohesin in the surrounding pericentromere

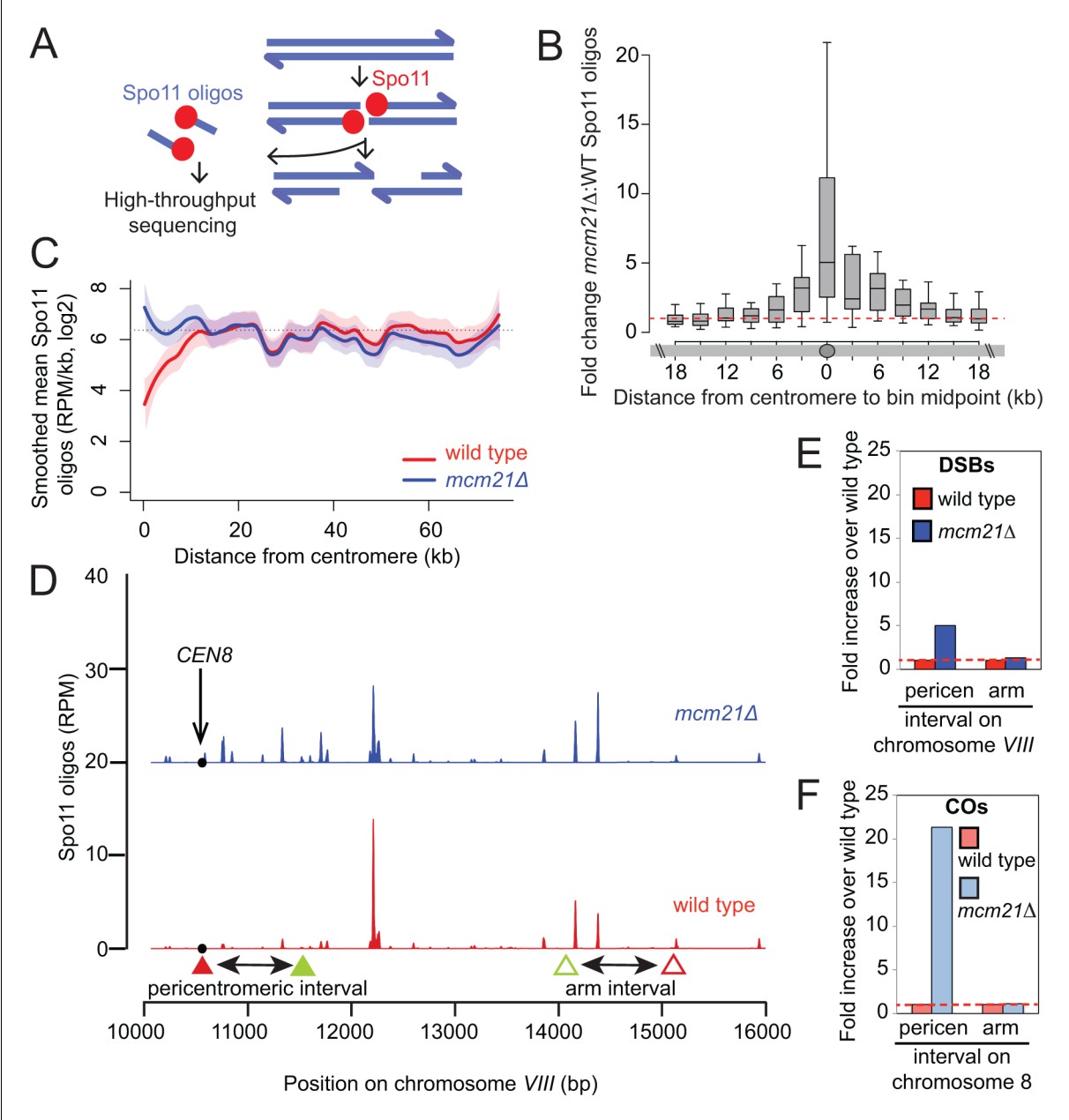

**Figure 3.** The kinetochore protects the centromere-proximal domain from DSBs. (A) Sequencing of Spo11-oligos allows DSBs to be mapped genome wide. (B) Fold change in average Spo11-oligo density (RPM per kb) in 3 kb segments in *mcm21Δ* cells compared to wild type (from **Zhu and Keeney (2015)**), over a 36 kb region surrounding all 16 centromeres. Boxes show median and interquartile range, whiskers enclose data points within 1.5 times the interquartile range; outliers are not shown. Red dashed line, fold change of one. (C) Mean Spo11 signal as a function of distance from the centromere. Spo11 oligo density within 500 bp bins starting from the centromere and moving up to 75 kb away, averaged across the 32 chromosome arms was determined. The horizontal dotted line indicates genome average. The red (wild type) and blue (*mcm21Δ*) lines indicate loess smoothing, and the shading indicates the 95% confidence interval. (D) DSBs in the region examined in the CO assay. Spo11 oligo counts smoothed with a 201-bp Hann window are shown. The black circle indicates the centromere, filled triangles indicate the midpoints of coordinates where RFP (red) and GFP (green) cassettes were targeted to for *CEN8* analysis in the live cell recombination assay; open triangles indicate the locations where the cassettes were targeted to for ARM8 analysis. (E and F) Fold change in the number of DSBs (E) or COs (F) in *mcm21Δ* vs. wild type within the same pericentromeric or arm intervals on chromosome 8 that were analyzed in the live cell recombination assay (**Figure 1B and C**). CO, crossover; DSBs, double strand breaks; RPM, reads per million mapped.

The following figure supplement is available for figure 3:

**Figure supplement 1.** Genome-wide view of meiotic recombination initiation in the *mcm21Δ* mutant.

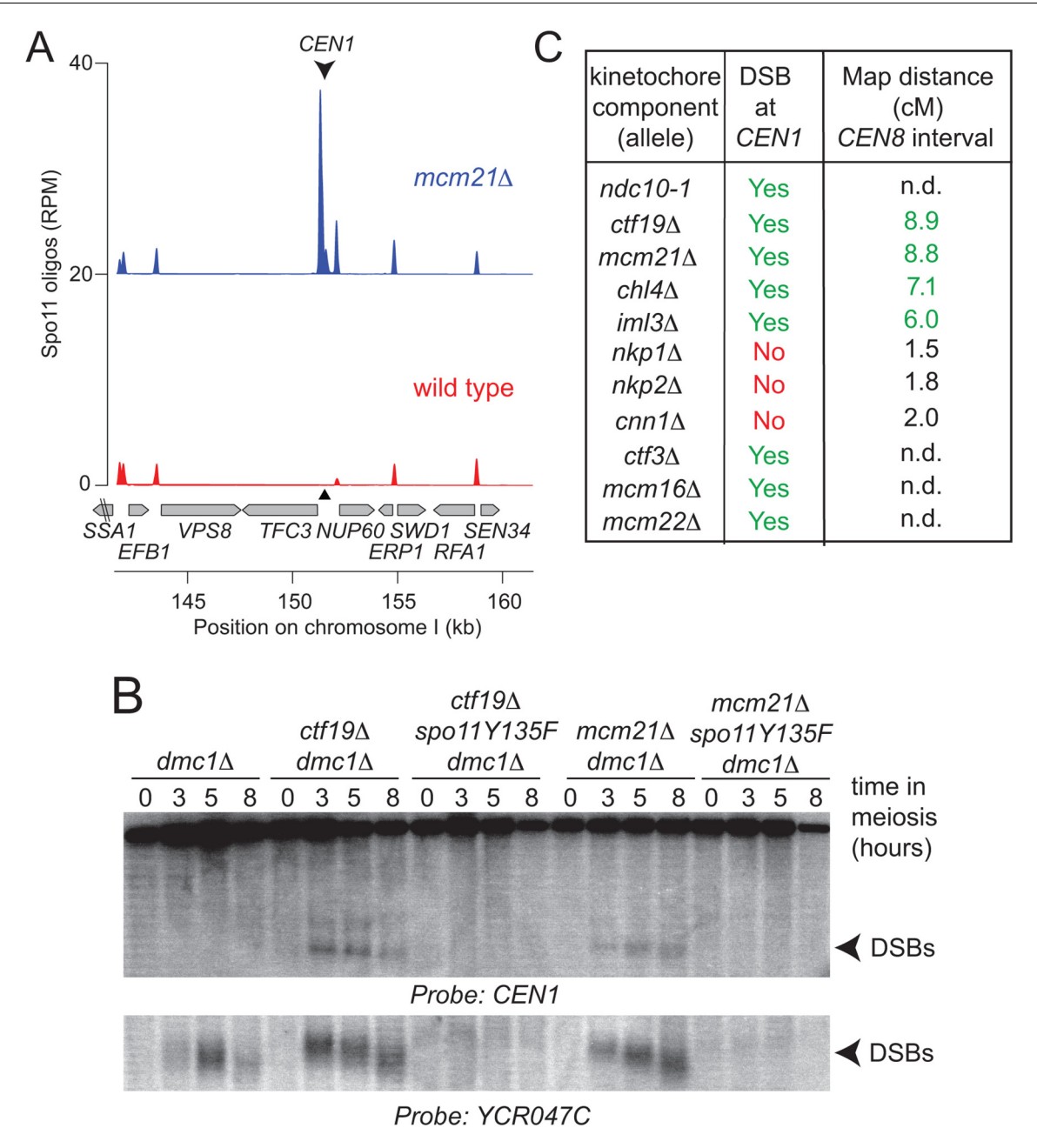

**Figure 4.** Analysis of DSB formation in Ctf19 complex mutants. (**A**) Appearance of a strong DSB hotspot proximal to *CEN1* in *mcm21Δ* cells. Spo11-oligo density in a 20 kb region surrounding the centromere of chromosome *I* in *mcm21Δ* (top) and wild type (bottom). Spo11-oligo counts (RPM) were smoothed with 201-bp Hann window. (**B**) Detection of *CEN1*-proximal DSBs in Ctf19 complex mutants by Southern blotting and their dependence on *SPO11* catalysis (using a catalytic dead mutant allele of *SPO11, spo11-Y135F*). Repair-deficient (*dmc1Δ*) cells were harvested at defined times after inducing sporulation ((t=0, 3, 5, 8 hr) and faster migrating DNA species (indicative of DSBs) were detected using a probe to *CEN1* or the control *YCR047C* locus. Arrowheads, Spo11-dependent DSBs. Strains used were GV48 (*dmc1Δ*), GV1912 (*dmc1Δ ctf19Δ*), GV2128 (*dmc1Δ ctf19Δ spo11-Y135F-HA*), GV2050 (*dmc1Δ mcm21Δ*) and GV2205 (*dmc1Δ mcm21Δ spo11-Y135F-HA*). (**C**) Summary of tested mutants and their importance for suppression of DSBs and inhibition of COs close to centromeres. n.d., not determined. CO, crossover; DSB, double strand break; RPM, reads per million mapped.

The following figure supplement is available for figure 4:

**Figure supplement 1.** Ctf19 complex components are required to prevent DSB formation close to *CEN1*.

(*Eckert et al., 2007*; *Fernius and Marston, 2009*; *Fernius et al., 2013*; *Ng et al., 2009*). This enrichment provides the basis for robust sister chromatid cohesion, the establishment of which is coupled to DNA replication in S-phase (*Eckert et al., 2007*; *Fernius and Marston, 2009*; *Fernius et al., 2013*; *Hinshaw et al., 2015*; *Ng et al., 2009*; *Uhlmann and Nasmyth, 1998*). Pericentromeric enrichment of the meiotic cohesin subunit, Rec8, at G2/prophase I also depended on the Ctf19 complex (*Figure 5A–D*). Since cohesin has been implicated in influencing meiotic DNA break formation and repair (*Ellermeier and Smith, 2005*; *Klein et al., 1999*; *Kugou et al., 2009*), we used the live cell recombination reporter assay to test the requirement for cohesin in preventing pericentromeric COs. To prevent pleiotropic phenotypes and sporulation failure associated with total cohesin loss in meiosis, we employed a mutation in the Scc4 subunit of the cohesin loader (*scc4-m35*), which in vegetative cells specifically abolishes pericentromeric cohesin enrichment (*Hinshaw et al., 2015*). In meiotic prophase, Rec8 levels were indeed reduced at centromeric and pericentromeric sites in *scc4-m35* cells. However, chromosomal arm sites were also affected (*Figure 5E*), suggesting that the *scc4-m35* mutations might influence cohesin loading at non-centromeric sites during meiosis. Nevertheless, *scc4-m35* cells underwent meiosis to produce spores, analysis of which revealed an increased frequency of pericentromeric (*Figure 5F*), but not chromosomal arm (*Figure 5G*) COs, though this increase was more modest than in the absence of Ctf19 complex subunits (*Figure 1B,C*). This finding supports the notion that pericentromeric cohesin enrichment by the Ctf19 complex contributes to the suppression of centromere-proximal COs.

We next asked whether the Ctf19 complex influences DSB patterns near centromeres by acting before and during S-phase, when it is known to promote the pericentromeric enrichment of cohesin. To test this, we used the "anchor away" system to selectively and conditionally deplete Ctf19 from the nucleus after S-phase (via timed addition of rapamycin (*Haruki et al., 2008*); *Figure 6A*, *Figure 6—figure supplement 1*). We reasoned that anchoring away Ctf19 before S phase (t=0) would prevent pericentromeric cohesin establishment, similar to a *ctf19△* strain (*Marston et al., 2005*; *Fernius et al., 2009*). In contrast, because cohesin establishment is coupled to DNA replication (*Uhlmann and Nasmyth, 1998*), addition of rapamycin after S phase (t=3) would allow cohesion establishment, thereby allowing us to test cohesin-independent functions of the Ctf19 complex (*Figure 6B*). First, we confirmed the successful removal of Ctf19-Frb-GFP from the centromere by addition of rapamycin either before (t=0) or after (t=3) DNA replication (*Figure 6C*). Next, we examined the effect on centromeric cohesin. Rec8 chromatin immunoprecipitation (ChIP) in prophase I cells revealed that centromeric cohesin levels were more greatly reduced by anchoring away Ctf19 before (t=0 h) rather than after DNA replication (t=3 h) (*Figure 6D*). Because this assay does not allow cohesin loaded before or after DNA replication to be distinguished, we sought to test the functionality of pericentromeric cohesin in the two conditions. Since only functional cohesin is expected to be retained at centromeres during anaphase I (*Klein et al., 1999*), we examined Rec8 on chromosome spreads in binucleate cells (*Figure 6E,F*). Although centromeric Rec8 was detected in only 4% of binucleate cells where Ctf19 was anchored away before DNA replication (Rapa t=0), centromeric Rec8 was observed in 48% of binucleate cells where Ctf19 was anchored away after DNA replication (Rapa t=3) (*Figure 6F*). Therefore, the presence of Ctf19 before and during S phase allows for the establishment of functional pericentromeric cohesin. Having established conditions that allowed us to uncouple pericentromeric cohesin establishment from post-S phase functions of the Ctf19 complex, we asked whether the role of the Ctf19 complex in suppressing pericentromeric DSBs is linked to its role in cohesin establishment during S phase. As expected, anchoring away Ctf19 before (Rapa t=0) DNA replication led to the appearance of *CEN1*-proximal DSBs with comparable timing and intensity to those of *ctf19△* cells (*Figure 6G–I*). DSBs were also observed following rapamycin addition at 3 hr, i.e. after DNA replication (*Figure 6H*). This suggests that the Ctf19 complex is required throughout meiotic prophase to prevent pericentromeric DSB formation and that it does so in a manner independent of its role in cohesin establishment.

To further test the requirement for cohesin in preventing pericentromeric DSB formation, we asked whether DSBs are increased near *CEN1* in cells defective for cohesin. Consistent with our findings above, *CEN1*-proximal DSBs were not observed in *rec8△* or *scc4-m35* cells, unless a Ctf19 complex component (Mcm21[CENP-O]) was also absent (*Figure 6J,K*). Thus, DSB inhibition near centromeres does not depend on cohesin, but requires the continuous presence of the Ctf19 complex. These findings provide an explanation for our observation that Ctf19 complex mutants exhibit a higher frequency of pericentromeric COs than *scc4-m35* mutant cells (*Figure 1C*, *Figure 5F*),

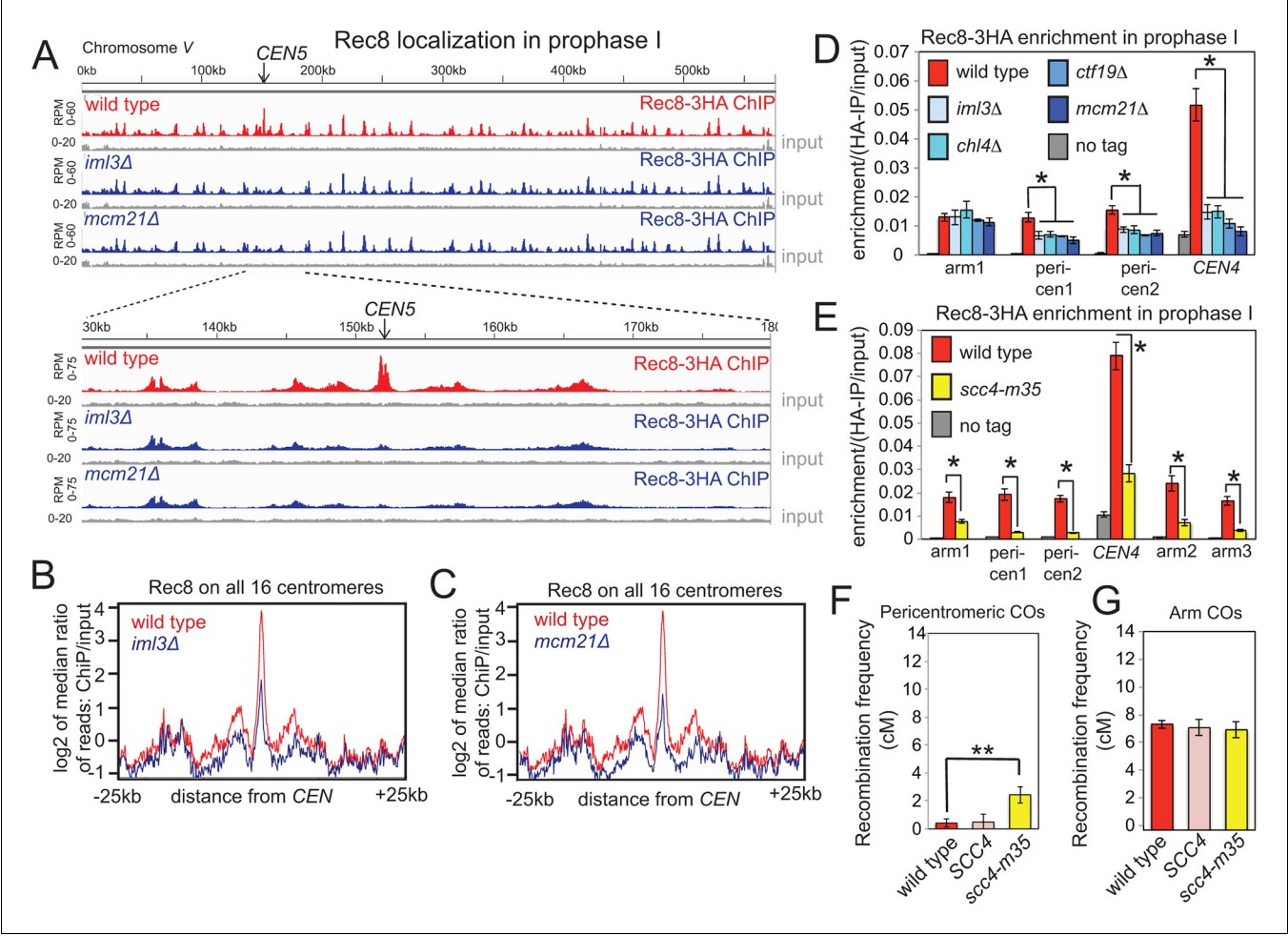

**Figure 5.** Ctf19-dependent cohesin enrichment prevents pericentromeric COs. (**A–D**), The Ctf19 complex enriches meiotic cohesin in the pericentromere during prophase I. (**A–C**) Wild type (AM4015), *iml3Δ* (AM4016) and *mcm21Δ* (AM13833) strains carrying *REC8-3HA* and *ndt80Δ* were harvested 5h after resuspension in sporulation medium and Rec8 association was analyzed by ChIP-Seq. Rec8 association with chromosome *V* and a close up of the 50 kb pericentromeric interval is shown (**A**). The median Rec8 level for all 16 pericentromeric regions is shown over a 25 kb region on each side of the centromere for *iml3Δ* (**B**) and *mcm21Δ* (**C**) compared to wild type. (**D**) Strains as in (**A**) together with *chl4Δ* (AM4017), *ctf19Δ* (AM20086) and a no tag control (AM11633) carrying *ndt80Δ* were arrested in prophase I by harvesting 5h after being induced to sporulate. The level of Rec8 at the indicated sites was determined by anti-HA ChIP-qPCR. Primer sets used corresponded to sites on the arm of chromosome *IV* (arm1, 2, 3), within the 20 kb pericentromere (pericen1, 2) or ~150bp from *CEN4* (*CEN4*) and sequences and coordinates are given in **Supplementary file 4B**. Error bars represent standard error (*n*=4 biological replicates for *iml3Δ*, *chl4Δ*, *ctf19Δ* and *mcm21Δ*; *n*=8 for no tag and wild type). *p<0.05, unpaired t test. (**E**) Chromosomal Rec8 levels are reduced in *scc4-m35* cells. Wild type (AM4015), *scc4-m35* (AM18211) cells and a no tag control (AM11633) carrying *ndt80Δ* were arrested in prophase I by harvesting 5 hr after being induced to sporulate. The level of Rec8 at the indicated sites was determined by anti-HA ChIP-qPCR. Error bars represent standard error (*n*=4 biological replicates). *p<0.05, paired t test. (**F, G**) Map distances (in cM) in the pericentromere (**F**) or a chromosome arm (**G**) interval in wild type, a control *SCC4* replacement strain, or the *scc4-m35* mutant were determined and their significance analysed as described in **Figure 1**. ChIP-Seq, chromatin immunoprecipitation with sequencing.

despite a comparable reduction in pericentromeric cohesin (**Figure 5D,E**). Loss of *MCM21* relieved DSB suppression over a domain (~6 kb; **Figure 3B**) that is much larger than the 125 bp where the kinetochore resides but smaller than the cohesin-rich pericentromere (~20 kb; **Figure 5A**). We therefore speculate that the large multi-subunit Ctf19 complex may exert DSB suppression near centromeres by altering local chromosome structure such that accessibility of DSB-promoting factors is prevented.

Our findings suggest that pericentromeric cohesin might provide a safeguarding mechanism to channel residual centromere-proximal DNA breaks towards repair pathways that do not promote

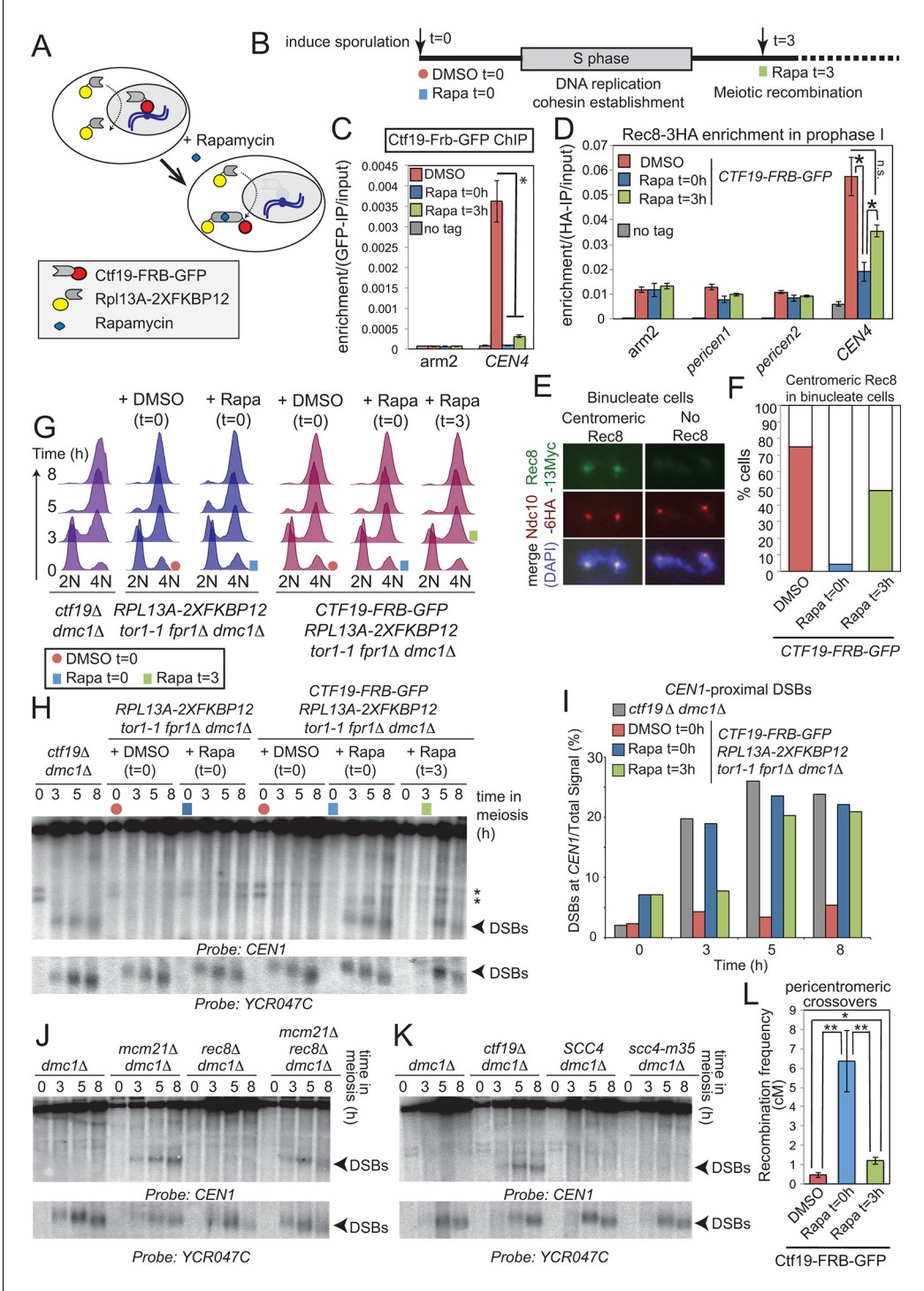

**Figure 6.** Pericentromeric cohesin does not prevent DSBs, but ensures their repair does not form COs. (**A–D**) Anchoring away Ctf19 after DNA replication and cohesion establishment is not sufficient to prevent the appearance of centromere-proximal DSBs. (**A, B**) Scheme of the anchor away system and experimental setup used to deplete Ctf19 during meiosis. (**C**) Addition of Rapamycin leads to Ctf19 removal from the pericentromere. Three cultures of strain AM18978 (*CTF19-FRB-GFP RPL13A-2XFKBP12 tor1-1 fpr1△ ndt80△ REC8-3HA*) were induced to sporulate. Either DMSO or Rapamycin were added to two of the cultures (t=0). Rapamycin was added 3 hr after inducing sporulation to the third culture (t=3). A fourth culture was a no tag *ndt80△* control (AM11633) to which DMSO was added. All cultures were harvested 5 hr after inducing sporulation (prophase I arrest) and Ctf19 levels were analyzed by anti-GFP ChIP-qPCR at the indicated sites. Error bars represent standard error (n=4

*Figure 6 continued on next page*

*Figure 6 continued*

biological replicates). *p<0.05, paired t-test. See *Figure 5* and *Supplementary file 4B* for details of primer sets used. (D–F) Anchoring away Ctf19 after DNA replication allows establishment of centromeric Rec8. (D) Cells treated as in (C) were processed for anti-HA ChIP-qPCR at the indicated sites. Error bars represent standard error (n=4 biological replicates). *p<0.05, paired t-test. (E and F) Three cultures of strain AM20138 (*CTF19-FRB-GFP NDC10-6HA pGAL-NDT80 pGPD1-GAL4(848)-ER RPL13A-2XFKBP12 tor1-1 fpr1△ ndt80△ REC8-3HA*) were resuspended in sporulation medium (t=0). DMSO or Rapamycin were immediately added to the first and second cultures (t=0), while the third culture received Rapamycin after 3 hr incubation (t=3). After 6 hr total, β-estradiol was added to release cells from the prophase I arrest, samples were harvested at 15 min intervals and chromosome spreads were prepared and stained with anti-HA and anti-Myc antibodies. (E) Examples of binucleate cells with centromeric or no Rec8. (F) Percentages of binucleate cells with centromeric Rec8 are shown for indicated conditions. (G–I) Timed depletion of Ctf19 to test DSB formation dependencies. (G) DNA replication is largely complete prior to anchoring Ctf19 away in cultures where Rapamycin was added at 3 hr. A control strain GV2367 (*RPL13A-2XFKBP12 tor1-1 fpr1△ dmc1△*) and equivalent experimental strain carrying *CTF19-FRB-GFP* (GV2354) were induced to undergo meiosis together with a *ctf19△ dmc1△* mutant (GV1912). Rapamycin or DMSO were added at the indicated times (red circles, addition of DMSO at t=0; blue squares, addition of Rapamycin at t=0; green squares, addition of Rapamycin at t=3) and samples were processed for FACS analysis (timepoints t=0, 3, 5, 8 hr). (H) Analysis of DSB formation in the experiment shown in (B). Southern blot shows that DSB formation close to *CEN1* occurs either when Ctf19 is anchored away early (t=0) or after DNA replication and cohesin enrichment (t=3). Red circles, addition of DMSO at t=0; blue squares, addition of Rapamycin at t=0; green squares, addition of Rapamycin at t=3; Arrowheads, Spo11-dependent DSBs; asterisks, cross-hybridizing species. (I) Quantification DSBs shown in (H). (J, K) Inhibition of centromere-proximal DSBs does not depend on cohesin. (J) *CEN1*-proximal DSBs are observed in *rec8△* cells only upon deletion of Ctf19 complex components. Strains GV48 (*dmc1△*), GV2050 (*dmc1△ mcm21△*), GV2403 (*dmc1△ rec8△*), GV2286 (*dmc1△ mcm21△ rec8△*) were analyzed by Southern blotting as described in *Figure 4B*. (K) Cohesin impairment does not allow *CEN1*-proximal DSB formation. Strains used were GV48 (*dmc1△*), GV1912 (*dmc1△ ctf19△*), GV2305 (*dmc1△ SCC4*) and GV2533 (*dmc1△ scc4-m35*). (L) Anchoring away Ctf19-FRB after DNA replication and cohesin establishment leads to only a modest increase in pericentromeric COs. Three cultures of strain AM19543 [carrying heterozygous pericentromeric RFP/GFP reporters separated by ~10 kb, homozygous chromosomal arm CFP reporters (*Figure 1B*), estradiol-inducible Ndt80 to allow release from prophase I arrest (*pGAL-NDT80 pGPD1-GAL4(848)-ER*) and the Ctf19 anchor away system (*CTF19-FRB RPL13A-2XFKBP12 tor1-1 fpr1△*)] were resuspended in sporulation medium (t=0). DMSO or Rapamycin were immediately added to the first and second cultures (t=0), while the third culture received Rapamycin after 3 hr incubation (t=3). All cultures were incubated for 6 hr total before being released from the prophase I arrest by addition of β-estradiol and tetrads were scored after incubation for 48 hr total. ChIP, chromatin immunoprecipitation; DMSO, dimethyl sulfoxide; DSBs, double strand breaks; FACS, fluorescence-activated cell sorting; qPCR, quantitative polymerase chain reaction.

The following figure supplement is available for figure 6:

**Figure supplement 1.** Anchoring Ctf19 away early in meiosis results in reduced sporulation efficiency and spore viability.

CO formation. If this is the case, DSBs arising after DNA replication and pericentromeric cohesion establishment would not be expected to give rise to pericentromeric COs. To test this idea, we anchored away Ctf19 either before (t=0) or after (t=3) DNA replication, both of which induce *CEN1*-proximal DSBs soon thereafter (*Figure 6H*) and measured CO formation using the live cell reporter assay (*Figure 6L*). Anchoring Ctf19 away before DNA replication (t=0) increased pericentromeric COs to a similar extent to *ctf19△* cells (compare *Figure 1B* and *Figure 6L*), as expected. However, anchoring Ctf19 away after DNA replication (t=3) led to a more modest increase in pericentromeric COs (*Figure 6L*), despite comparable levels of DNA break formation in these cells (*Figure 6H,I*). Therefore, pericentromeric cohesin acts at a step after DSB formation to direct repair, probably through a pathway avoiding the homolog, to ensure CO suppression near centromeres. Because the cohesin complex is known to promote inter-sister recombination in mitosis and meiosis (*Covo et al., 2010*; *Ellermeier and Smith, 2005*; *Klein et al., 1999*; *Kugou et al., 2009*; *Sjogren and Nasmyth, 2001*) the simplest explanation is that increased pericentromeric cohesin shunts meiotic DSBs into inter-sister-specific recombinational repair, although we cannot rule out pericentromere-specific, cohesin-dependent activation of alternative repair pathways.

## Zip1 prevents pericentromeric COs and promotes centromere pairing through separate mechanisms

Rec8 globally influences the localization of the synaptonemal complex (SC) component Zip1 along chromosomes (*Chuong and Dawson, 2010*; *Brar et al., 2009*). Because Zip1, like cohesin (*Kim et al., 2010*) has been suggested to confer a bias on DSBs to be repaired from the sister chromatid, rather than the homolog, and because loss of Zip1 increases pericentromeric COs (*Figure 1B*), but not DSBs (*Chen et al., 2008*; *Figure 7—figure supplement 1*), we reasoned that a critical role of pericentromeric cohesin might be Zip1 recruitment. Consistently, ChIP-qPCR indicated that Zip1 localization was impaired in *scc4-m35* (*Figure 7A*) and Ctf19 complex mutants (*Figure 7B*). Furthermore, analysis of spread meiotic chromosomes revealed that the "dotty" Zip1 localization pattern, representative of centromeres, was rarely observed in Ctf19 complex mutants (*Figure 7C*). Unexpectedly, "full" Zip1 localization along chromosomes was also impaired in Ctf19 complex mutants. Although the underlying reasons for this are currently unclear, possible explanations are delayed G2/prophase progression and/or a requirement for pericentromeric Zip1 in producing the "full" Zip1 tracts observed in cytological analyses. Nevertheless, ChIP-Seq of prophase I-arrested cells confirmed that Ctf19 complex components are specifically required for Zip1 association with core centromeres and the pericentromere (*Figure 7D–F*). We measured comparable pericentromeric CO frequencies of 2.4 and 2.0 cM in *scc4-m35* and *zip1△* cells, respectively (*Figure 1B*, *Figure 5F*) suggesting that pericentromeric cohesin establishment by the Ctf19 complex directs Zip1 association to suppress pericentromeric COs, although we do not rule out Zip1-independent functions of cohesin in CO suppression.

Centromeric Zip1 mediates homology-independent pairing of homologous chromosomes early in meiotic prophase; the subsequent conversion to homologous pairing requires Spo11 (*Tsubouchi and Roeder, 2005*; *Tsubouchi et al., 2008*). The Ctf19 complex was required for the centromeric localization of Zip1 in *spo11△* cells (*Figure 8A*) and homology-independent centromere coupling (*Figure 8B*) but not for the *SPO11*-dependent transition to homologous pairing (*Figure 8C,D*). To test whether Zip1 exerts its role in suppression of pericentromeric COs through homology-independent centromere coupling, we analyzed the synaptonemal complex assembly-proficient but centromere coupling-defective *zip1-S75E* mutant (*Falk et al., 2010*), using the live cell recombination reporter assay. Pericentromeric COs were not significantly increased in the *zip1-S75E* mutant (*Figure 8E,F*), indicating that Zip1 suppresses pericentromeric COs independently of its role in centromere coupling.

## Discussion

Centromere-proximal CO recombination is a major risk factor for meiotic chromosome segregation and developmental aneuploidy and is suppressed in species with diverse centromere organization. The data presented here establish the highly conserved kinetochore as a major factor responsible for setting up a repressive environment for crossover recombination in the pericentromere during meiosis (*Figure 9*). In cells lacking the Ctf19 kinetochore sub-complex, pericentromeric CO formation drastically increases. We uncover multi-layered Ctf19 complex-dependent suppression of CO formation acting at both the level of meiotic DSB formation and at the level of recombinational repair template choice. This centrally important role of the kinetochore in preventing potentially detrimental crossovers nearby defines a further mechanism by which it safeguards genome stability.

### The Ctf19 complex suppresses centromere-proximal DNA breaks

The placement of meiotic DSBs is influenced by factors acting on different levels of chromosome and chromatin organization. On a global scale, the assembly of a meiotic chromosome axis dictates the spatial distribution of the meiotic DSB machinery along chromosomes (*Kim et al., 2010*; *Kugou et al., 2009*; *Pan et al., 2011*; *Panizza et al., 2011*). On a smaller scale, genome organization and histone modifications have been shown to allow Spo11 activity (reviewed in *de Massy (2013)*). Spo11-dependent DSB formation is strongly associated with chromatin regions that are enriched for histone H3 Lysine 4 (H3K4) methylation (reviewed in *de Massy (2013)*) and in budding yeast, these modifications are found near transcriptional start sites (*Tischfield and Keeney, 2012*). Indeed, within the budding yeast genome, Spo11 prefers to cleave DNA in open, nucleosome-free

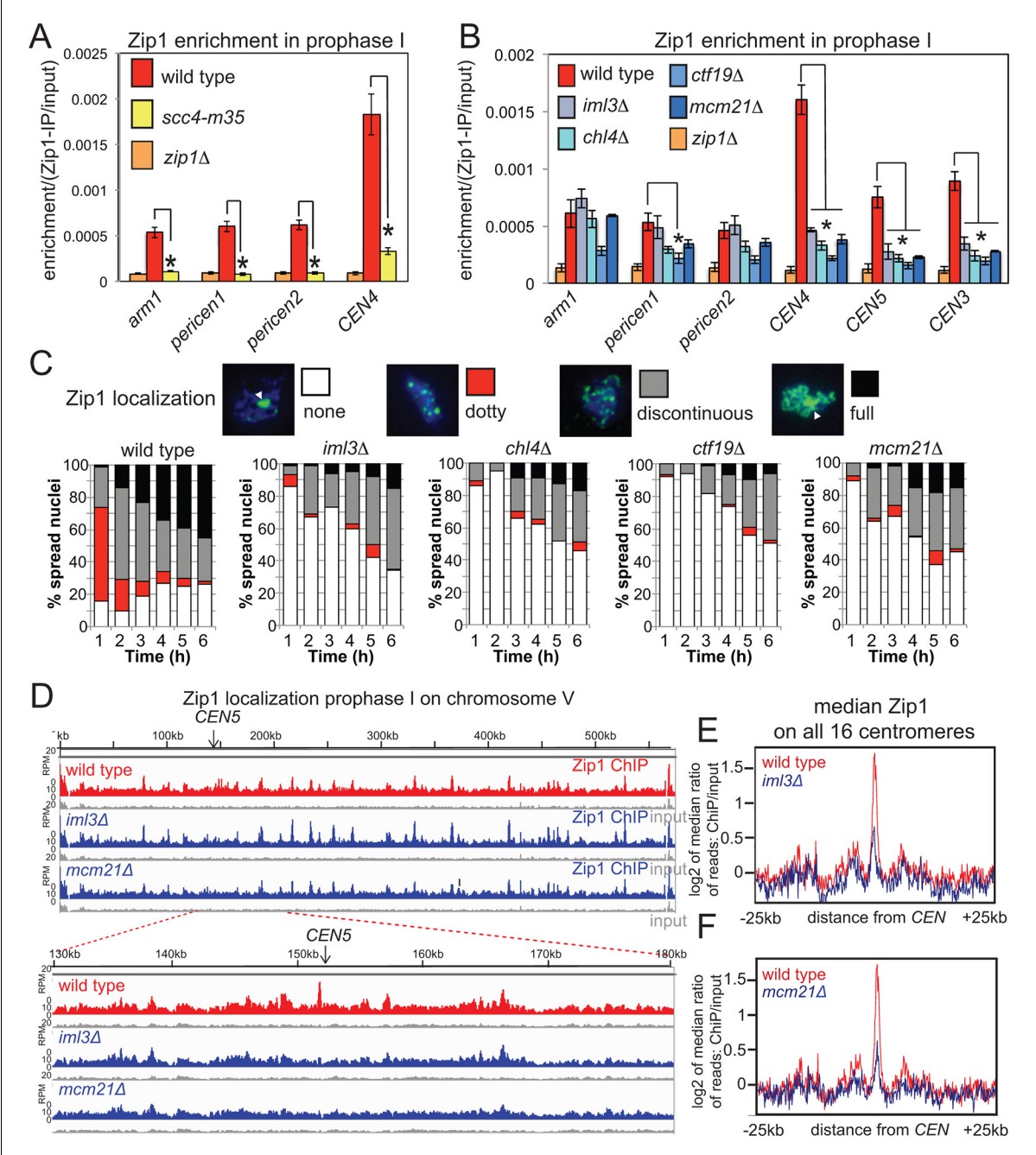

**Figure 7.** Cohesin enables centromeric Zip1 recruitment. (**A**) Zip1 enrichment on chromosomes is reduced in *scc4-m35* mutants. Wild type (AM11633), *scc4-m35* (AM18881) and *zip1△* (AM10913) cells carrying *ndt80△* were induced to sporulate and harvested after 5 hr (prophase I) arrest for anti-Zip1 ChIP-qPCR. Error bars represent standard error (*n*=4 biological replicates). p<0.05, paired t test. See **Figure 5** and **Supplementary file 4B** for details of primer sets used. (**B—F**) The Ctf19 complex is required for Zip1 localization at centromeres. (**B**) ChIP-qPCR analysis of Zip1 localization in prophase I in Ctf19 complex mutants. Wild type (AM11633), *iml3△* (AM10686), *chl4△* (AM10658), *ctf19△* (AM10660), *mcm21△* (AM10664) and *zip1△* (AM10913) cells carrying *ndt80△* were induced to sporulate and harvested after 5 hr for anti-Zip1 qPCR. Error bars represent standard error (*n*=3 biological replicates). p<0.05, paired t test. See **Figure 5** and **Supplementary file 4B** for details of primer sets used. (**C**) Analysis of Zip1 localization on chromosome spreads as cells progress into prophase I. Examples of Zip1 localization on chromosome spreads are shown with Zip1 in green and DNA in blue. Categories of Zip1 localization were scored in 100 spread nuclei at each of the indicated times after resuspension in sporulation medium. Strains used carried *NDC10-6HA, pGAL-NDT80, pGPD1-GAL4-ER* and were AM8769 (wild type), AM8772 (*iml3△*), AM8770 (*chl4△*), AM9049 (*ctf19△*) and AM8861 (*mcm21△*). (**D–F**) ChIP-Seq analysis of Zip1 localization during prophase I in wild type, *iml3△* and *mcm21△* strains carrying *ndt80△* (B) and harvested 5 hr after resuspension in sporulation medium. (**D**) Zip1 localization along chromosome *V* is shown as an example with the 50 kb region around the centromere

*Figure 7 continued on next page*

*Figure 7 continued*

amplified. (E, F) Median Zip1 localization over a 50 kb domain surrounding all 16 centromeres is shown compared to wild type for *iml3△* (E) and *mcm21△* (F). ChIP, chromatin immunoprecipitation; qPCR, quantitative polymerase chain reaction.

The following figure supplement is available for figure 7:

**Figure supplement 1.** Zip1 does not induce pericentromeric DSB formation.

regions that are most often found in active, divergent promoters (*Blitzblau et al., 2007*; *Pan et al., 2011*).

Superimposed on these global determinants of the DSB landscape are specific and spatial controls, which create zones of inhibition within at-risk genomic regions such as telomeres, repetitive DNA arrays and centromeres (reviewed in *de Massy (2013)*). We find that the kinetochore actively minimizes DSB formation within a region of ~6 kb, surrounding all budding yeast centromeres. The underlying genome organization of these regions is not obviously different from the rest of the

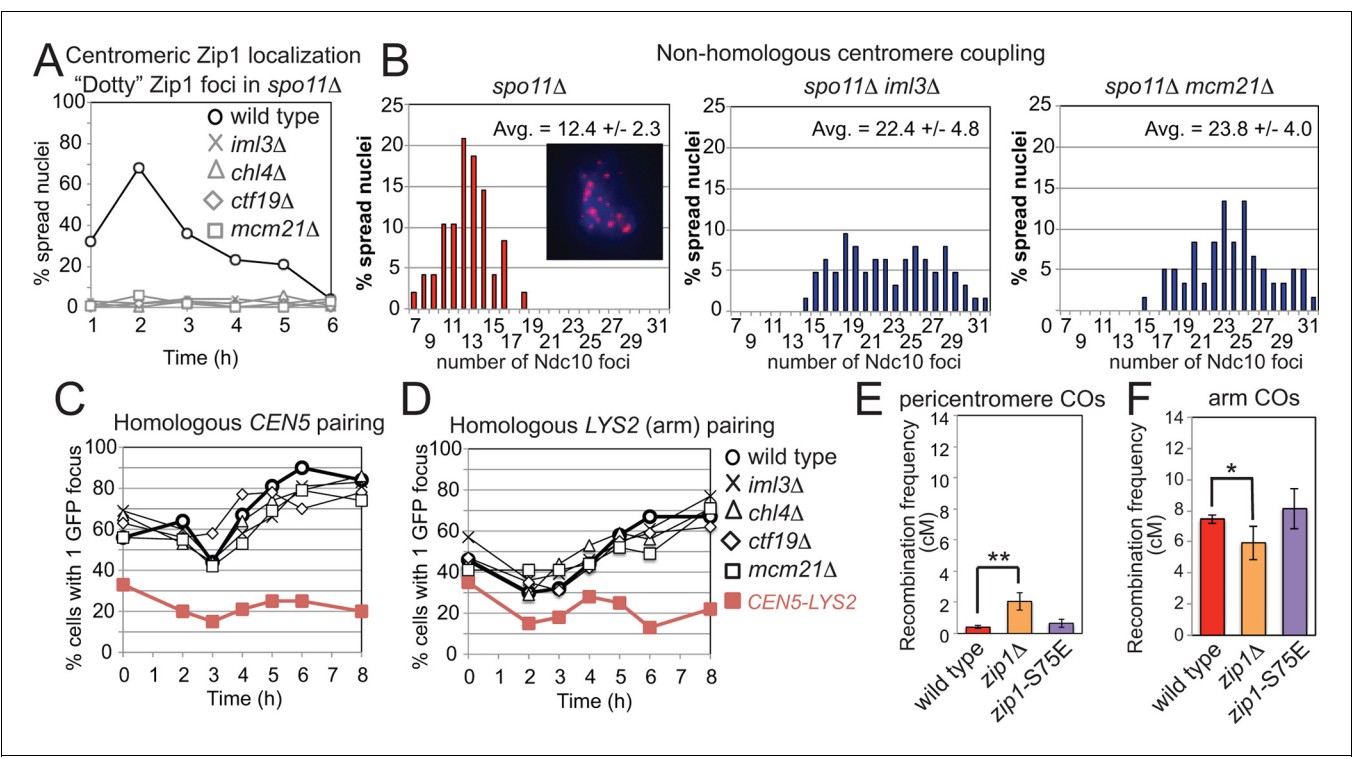

**Figure 8.** Recruitment of Zip1 to centromeres by the Ctf19 complex independently promotes centromere coupling and suppression of pericentromeric COs. (A, B) Analysis of Zip1 and kinetochore foci (Ndc10-6HA) on chromosome spreads in *spo11△* cells progressing into meiotic prophase I. The percentages of cells with "dotty" Zip1 foci, representing kinetochores, were scored in 100 spread nuclei at each of the indicated times after resuspension of strains *spo11△* (AM9018), *spo11△ iml3△* (AM9288), *spo11△ chl4△* (AM9287), *spo11△ ctf19△* (AM9017) and *spo11△ mcm21△* (AM8861) in sporulation medium (A). The number of Ndc10 kinetochore foci per spread nucleus was scored in the indicated strains at the 6 hr time point with the average and standard deviation indicated. n=48 (*spo11△*), 63 (*spo11△ iml3△*) and 60 (*spo11△ mcm21△*). (B) The inset shows Ndc10-6HA staining (red) in an example nucleus. DNA is shown in blue. (C) The Ctf19 complex is not required for *SPO11*-dependent homologous pairing. Pairing of homozygous *CEN5*-GFP (C) or *LYS2*-GFP (D, arm) foci was scored for 100 cells at each of the indicated times after resuspension in sporulation medium in *ndt80△* cells. Heterologous *CEN5-GFP* and *LYS2-GFP* foci labels were used as a control for spurious interactions (strain AM12823, red filled squares). Strains used in (C) were AM12829 (wild type), AM13348 (*iml3△*), AM12466 (*chl4△*), AM13346 (*ctf19△*) and AM12837 (*mcm21△*). Strains used in (D) were AM12469 (wild type), AM12978 (*iml3△*), AM12980 (*chl4△*), AM12831 (*ctf19△*) and AM12825 (*mcm21△*). A representative experiment is shown. (E, F) The centromere-coupling function of Zip1 is separable from its role in suppression of recombination in the pericentromere. Recombination frequency in the pericentromere (E) or chromosomal arm (F) interval on chromosome *VIII* for the centromere-coupling defective *zip1-S75E* mutant is shown together with data for wild type and *zip1△* reproduced from *Figure 1B,C*. COs, crossovers.

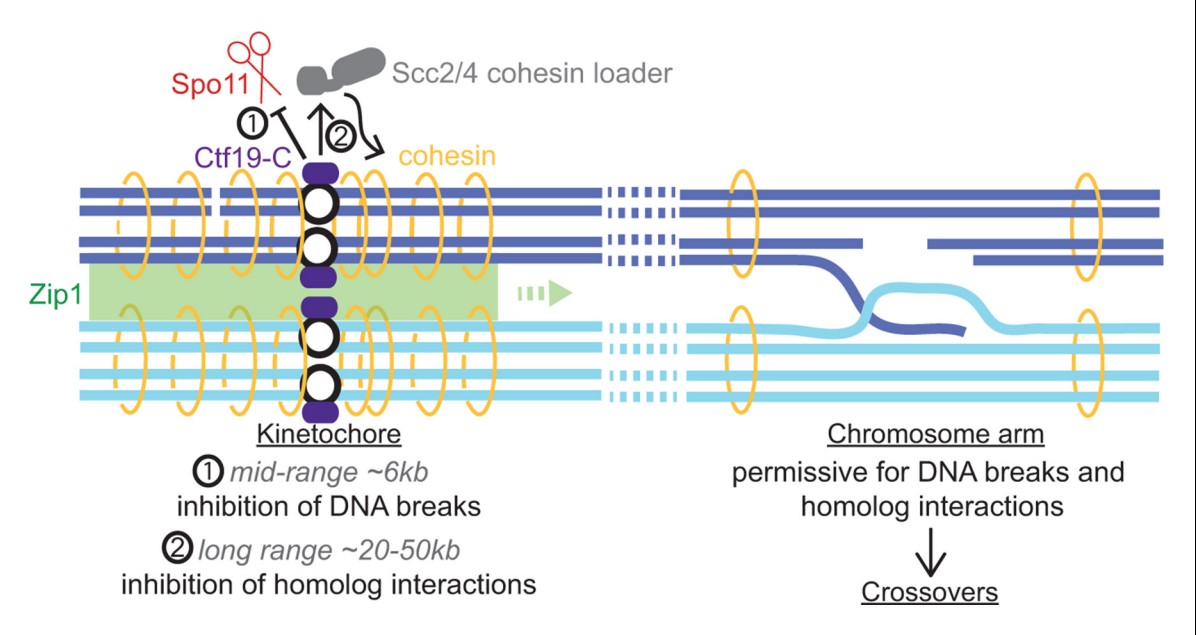

**Figure 9.** Schematic diagram indicating dual repression of pericentromeric recombination by the Ctf19 kinetochore sub-complex. (1) Mid-range (~6 kb) suppression of DSBs. (2) Long range (~20-50 kb) inhibition of COs by cohesin and Zip1, directing repair of rare centromere-proximal breaks to the sister chromatid to avoid deleterious pericentromeric inter-homolog chiasmata.

genome, harboring a similar density of genes and regulatory elements, which would ordinarily be expected to contain several DSB-permissive regions, were they not located close to a centromere. Indeed, in the absence of the Ctf19 kinetochore sub-complex, DSB formation is strongly increased, and these DSBs have features typical of preferred DSB sites genome-wide. Accordingly, one of the strongest pericentromeric DSBs that we identified here, at *CEN1*, falls within a divergent promoter (of the genes *TFC3* and *NUP60*; *Figure 4A*). The Ctf19 complex suppresses DSB formation surrounding all centromeres, though to different extents depending on the chromosome (*Figure 3—figure supplement 1*), suggesting that it overcomes the intrinsic features of the chromatin organization in the pericentromere to dampen DSB activity. This could be achieved by locally shaping a higher order chromosome organization and/or by influencing the recruitment of meiotic chromosome axis components. Intriguingly, the chromosomal region protected from DSBs (~6 kb) extends far beyond the binding of the Ctf19 complex, which is restricted to the 125 bp point centromere, as defined by the assembly of a nucleosome containing the centromere-specific histone variant Cse4$^{CENP-A}$ (*Biggins, 2013*). Therefore, the Ctf19 complex protects a region about 50 times larger than that occupied by the centromeric nucleosome from DSBs, suggesting that the effect is not merely due to a local disturbance of Cse4$^{CENP-A}$. A precedent for the idea that the Ctf19 complex can exert long-range effects on the pericentromere through its role in driving cohesin loading at the centromere to enrich the surrounding 20–50 kb is well-documented (*Eckert et al., 2007*; *Fernius and Marston, 2009*; *Fernius et al., 2013*; *Ng et al., 2013*). However, we found that the Ctf19 complex must prevent DSB formation independently of its role in promoting cohesin enrichment within pericentromeres. Cohesin enrichment was neither sufficient nor required to prevent DSBs near *CEN1*. Furthermore, DSB suppression acts on shorter chromosomal distances (~6 kb), than the ~20–50 kb sized regions within which the Ctf19 complex influences cohesin recruitment. Taken together, these findings suggest the existence of an additional, distinct Ctf19 complex-dependent effect on meiotic DSB formation within regions adjacent to centromeres. By analogy to the effect on pericentromeric cohesin enrichment, we speculate that the Ctf19 complex may enable the centromeric recruitment of factors that alter chromatin organization in the surrounding region.

## Cohesin recruitment in the pericentromere by the Ctf19 complex diverts DNA repair pathways away from the homolog to prevent CO formation

Minimizing the initiating event of meiotic recombination, DSB formation, is an efficient way to shield against unwanted pericentromeric CO formation. However, the prevention of pericentromeric DSBs by the Ctf19 complex is not absolute, as DSBs are observed near centromeres in wild type cells, although at reduced levels (*Blitzblau et al., 2007*; *Buhler et al., 2007*; *Pan et al., 2011*). We found that DSBs that escape the repressive control of the Ctf19 complex are diverted from repair pathways that would lead to potentially deleterious CO formation by cohesin, which is established at high levels within pericentromeres. These observations are in agreement with previous conclusions that pericentromeric CO formation is more strongly suppressed than DSB formation (*Blitzblau et al., 2007*; *Buhler et al., 2007*; *Chen et al., 2008*; *Pan et al., 2011*).

We found that forced removal of the Ctf19 complex from kinetochores after S-phase triggered increased DSB formation, but led to only a relatively modest increase in CO formation. Conversely, a cohesin-loader mutant (*scc4-m35*) that disrupts the proper establishment of cohesin showed increased levels of CO formation, whereas no increased DSB formation could be detected. Thus, DSBs that occur within the pericentromeric regions when high levels of cohesion are present are shunted away from inter-homologous recombinational repair (that eventually can yield a CO or NCO). Taking into account the established roles for cohesin in promoting inter-sister repair (*Covo et al., 2010*; *Ellermeier and Smith, 2005*; *Klein et al., 1999*; *Kugou et al., 2009*; *Sjogren and Nasmyth, 2001*), we consider it is most likely that these breaks will preferentially be repaired using the sister chromatid as a repair template.

How would the enrichment of cohesin minimize inter-homologue repair? One possibility is that high levels of cohesin turn the sister chromatid within pericentromeres into a preferred repair template. Indeed, it has been shown that template choice (sister or homologue) for DSB repair during budding yeast vegetative growth is dictated by the levels of cohesin (*Covo et al., 2010*). Globally, on meiotic chromosomes, repair of DSBs is normally biased towards use of the homologous chromosome instead of the sister chromatid to ensure at least one COs is formed to link each homolog pair (*Hollingsworth, 2010*; *Humphryes and Hochwagen, 2014*). In budding yeast, this homologue bias is established by the Red1/Hop1/Mek1 signaling axis (*Hong et al., 2013*; *Kim et al., 2010*; *Niu et al., 2007*; *Niu et al., 2005*). This signaling pathway is thought to locally antagonize cohesin to enable the use of the more distant homologous chromosome as a repair template (*Hong et al., 2013*; *Kim et al., 2010*). Potentially, the Red1/Hop1/Mek1 system is not capable of counteracting cohesin within pericentromeres, where much higher levels of cohesin are present as compared to elsewhere in the genome. Alternatively, one could envision that, within pericentromeric regions, Red1/Hop1/Mek1 are incapacitated via active inhibition or removal.

Our findings point to a second mechanism through which cohesin steers repair of meiotic DSBs away from inter-homologue repair: by promoting the local recruitment of Zip1. Zip1 has previously been implicated in preventing pericentromeric CO and NCO repair, and was suggested to promote inter-sister repair (*Chen et al., 2008*). We found that proper recruitment of Zip1 to the pericentromere requires the Ctf19 complex and kinetochore-targeted Scc2/4, consistent with previous studies demonstrating a requirement for cohesin in Zip1 recruitment (*Chuong and Dawson, 2010*; *Brar et al., 2009*). Zip1 performs specific functions at centromeres during prophase, which include non-homologous centromere-coupling (*Tsubouchi and Roeder, 2005*; *Tsubouchi et al., 2008*), repression of pericentromeric CO formation (*Chen et al., 2008*) and the bi-orientation of homologous chromosomes during meiosis I (*Gladstone et al., 2009*; *Newnham et al., 2010*). Here we have shown that the Ctf19 complex, presumably through its role in directing pericentromeric cohesin enrichment, enables the dedicated recruitment of a centromere-localized pool of Zip1 to perform these specialized functions.

Overall, we conclude that the kinetochore, and specifically the Ctf19 complex, promotes the establishment of an inter-homologue recombination-suppressed zone surrounding centromeres. To do so, the Ctf19 complex recruits high levels of cohesin within pericentromeres, which in turn triggers efficient recruitment of the synaptonemal complex component Zip1, effectively leading to strong, local inhibition of inter-homologue directed repair.

## A novel role for the kinetochore during meiotic G2/Prophase

CO formation in the vicinity of centromeres negatively influences meiotic chromosome segregation in diverse organisms, and is associated with the incidence of Trisomy 21, or Down's syndrome, in humans (*Hassold and Hunt, 2001*; *Koehler et al., 1996*; *Rockmill et al., 2006*). Mechanistically, pericentromeric CO formation has been suggested to lead to a local disturbance of sister chromatid cohesion, which could lead to precocious separation sister chromatid, causing meiosis II non-disjunction and aneuploidy (*Rockmill et al., 2006*). While meiosis II mis-segregation is a characteristic feature of Ctf19 complex mutants (*Fernius and Marston, 2009*), the impact of pericentromeric COs on this phenotype is currently unclear because of the requirement for this complex for proper pericentromeric cohesion.

The kinetochore is a sophisticated machine that couples chromosomes to microtubules and drives their segregation in meiosis and mitosis. The kinetochore also serves as a signaling platform that monitors and responds to the state of kinetochore–microtubule attachment in the context of the cell cycle. During meiosis, in addition to the canonical events that also take place during mitosis, the kinetochore takes on additional roles to bring about the specialized segregation pattern. First, sister chromatids need to attach to microtubules that emanate from the same pole in a mono-oriented fashion. Second, sister chromatid cohesion at pericentromeres needs to be protected from removal during meiosis I. In both cases, the kinetochore controls these events by coordinating the recruitment of specific protein complexes (i.e. monopolin and Sgo1/PP2A, respectively) (reviewed in *Duro and Marston (2015)*). Our data add a hitherto unknown, additional level of functionality to the kinetochore in meiosis, in which it impacts meiosis-specific CO formation by influencing both meiotic DSB formation and recombinational break repair. As such, it effectively prevents the formation of unwanted chiasma within pericentromeres. Finally, we note that our data might also provide an additional rationale for why much higher cohesin levels need to be established around centromeres, namely to minimise CO formation near centromeres.

The Ctf19 complex is a generally conserved component of eukaryotic kinetochores. Suppression of meiotic CO recombination within pericentromeres is a widespread feature of meiotic recombination in many diverse organisms, whether or not their centromeres are surrounded by large blocks of heterochromatinized DNA. We therefore suggest that Ctf19 complex-driven suppression of meiotic CO formation serves as a universal component of the mechanisms that shape the meiotic recombination landscape in order to promote the faithful propagation of the genome from generation to the next.

## Materials and methods

### Yeast strains and plasmids

Yeast strains used in this study are derivatives of SK1 and genotypes are given in *Supplementary file 4A* except for the strains used for analysis of recombination genome-wide where AM15182 and AM15183 haploid strains derived from YJM789 and S96, respectively, were used. Standard techniques were used to generate gene deletions, promoter replacements and epitope-tagged proteins. *zip1-S75E* was described in *Falk et al. (2010)*. *scc4-m35* and *SCC4::HIS3* were generated in SK1 as described by *Hinshaw et al. (2015)*. *SPO11-6HIS-3FLAG-loxP-KanMX-loxP* was provided by K. Ohta (*Kugou et al., 2009*) and *DSN1-6HIS-3FLAG* was described in *Sarangapani et al., (2014)*. For the anchor away system, parental SK1 strains were generated as described by *Haruki et al. (2008)* harbouring *tor1-1, fpr1Δ*, and *RPL13A-2xFKBP12*. *CTF19-FRB-GFP* was made using standard polymerase chain reaction (PCR)-based transformation using pFA6a-FRB-GFP-KanMX6 (GVp584) as a template (*Haruki et al., 2008*). Prophase I block-release experiments used strains carrying *pGAL-NDT80 pGPD1-GAL4(848)-ER* (*Benjamin et al., 2003*).

For the live cell recombination reporter assay, *pYKL050c-CFP/RFP/GFP** constructs (*Thacker et al., 2011*) were integrated at specific loci. Plasmids AMp1005 and AMp1048 were generated by cloning an ~500 bp region corresponding to SGD coordinates 115024–115572 and 150521–151070 into pSK726 and pSK691, respectively. *pYKL050-CFP* was introduced at the *THR1* locus by integration of plasmid pSK695 and *pYKL050-RFP* was integrated at the *CEN8* locus by integration of plasmid pSK693 or at *SGD* coordinates 150521–151070 by integration of plasmid

AMp1048. *pYKL050c-GFP** was introduced at the *ARG4* locus by integration of the plasmid pSK729 and at *SGD* coordinates 115024–115572 locus by integration of plasmid AMp1005.

## Growth conditions

Diploid yeast strains were placed on Yeast peptone glycerol (YPG) agar plates (1% yeast extract, 2% Bacto peptone, 2.5% glycerol, and 2% agar) and grown for 16 hr at 30°C before transferring to YPD 4% agar plates (1% yeast extract, 2% Bacto peptone, 4% glucose, and 2% agar) and incubated for 24 hr at 30°C. Strains were inoculated in YPD media (1% yeast extract, 2% Bacto peptone, and 2% glucose) and cultured for 24 hr before being transferred to YPA (1% yeast extract, 2% Bacto peptone, and 1% potassium acetate) or BYTA (1% yeast extract, 2% Bacto tryptone, 1% potassium acetate, 50 mM potassium phthalate) at an $OD_{600}$ = 0.2–0.3 for ~16 hr. Cells were washed once with sterile distilled water and re-suspended in SPO media (0.3% potassium acetate, pH 7) at an $OD_{600}$ = 1.8–1.9; t=0. Cells were incubated at 30°C for the duration of the experiment. Prophase I block-release experiments were performed as described by *Carlile and Amon (2008)*.

## Western blotting

Western blotting was performed as previously described (*Clift et al., 2009*), with the exception that proteins were visualized using a fluorophore-conjugated antibody and the Odyssey system (LI-COR Biosciences, Lincoln, Nebraska). To visualise 3HA-Iml3, 3HA-Dsn1 and 3HA-Mtw1, mouse anti-HA (12CA5, Roche, Basel, Switzerland) was used at a dilution of 1:1000 and anti-mouse IRDye 800CW (LI-COR Biosciences) at a dilution of 1:10,000. To visualise Pgk1 (loading control), rabbit anti-Pgk1 (Marston lab stock) and anti-rabbit IRDye 680RD (LI-COR Biosciences) were used at a dilution of 1:10000.

## Anchor-Away technique

Proteins tagged with the FKBP12-rapamycin-binding (FRB) domain of mTOR1 were depleted from the nucleus by Rpl13A-2xFKBP12 upon addition of rapamycin to a final concentration of 1 µM, as previously described (*Haruki et al., 2008*).

## Chromatin immunoprecipitation

ChIP-qPCR and ChIP-Seq were performed as described in *Verzijlbergen et al. (2014)* using mouse anti-HA (12CA5, Roche), rabbit anti-Zip1 (Santa Cruz Biotechnology, Dallas, Texas), mouse anti-FLAG (Mono M2, Sigma Aldrich, St Louis, Missouri) or mouse anti-GFP (Sigma Aldrich). Primers used for qPCR analysis are given in *Supplementary file 4B*.

## Analysis of ChIP-Seq data

ChIP-Seq samples were analysed on a HiSeq2000 instrument (Illumina, San Diego, California) by the EMBL Core Genomics Facility (Heidelberg, Germany). Using BWA (Version: 0.7.5a-r405) (*Li and Durbin, 2010*), single reads were mapped to the sacCer3 reference genome. Duplicate reads were removed for parallel analysis using SAMtools (Version 1.2) (*Li et al., 2009*). The data shown were normalized to the number of reads per million of total mapped reads. The total mapped reads were established after any processing. Additional scripts for processing data around the pericentromeres can be found at https://github.com/AlastairKerr/Vincenten2015. ChIP-Seq data sets have been deposited with the NCBI Gene Expression Omnibus under the accession number GSE70032.

## Microscopy

Chromosome spreading was performed as described previously (*Loidl et al., 1998*). Ndc10-6HA was detected using a mouse anti-HA antibody (Mono HA.11, Covance, Princeton, New Jersey) at 1:250 dilution and an anti-mouse Cy3 antibody (Jackson ImmunoResearch) at 1:300 dilution. Rec8-13 Myc was detected with a rabbit anti-MYC antibody (Gramsch Laboratories) and an anti-rabbit fluorescein isothiocyanate (FITC)-conjugated antibody (Jackson Immunoresearch, West Grove, Pennsylvania), both at 1:300 dilution. Zip1 was detected with a rabbit anti-Zip1 antibody (Santa Cruz Biotechnology) at 1:500 dilution and an anti-rabbit FITC antibody (Jackson ImmunoResearch) at 1:300 dilution. Chromosome spread samples were analysed on a DeltaVision Elite system (Applied Precision, Isaaquah, Washington) using an inverted Olympus IX-71 microscope with a 100x

UPlanSApo NA 1.4 oil lens. Images were acquired using the Photometrics Cascade II EMCCD camera. The camera, shutters, and stage were operated through SoftWorx software (Applied Precision). Samples for studying GFP-labelled chromosomes were prepared as described previously (*Klein et al., 1999*). For the recombination and GFP-labelled chromosome assays, microscopy analysis was performed using a Zeiss Axioplan 2 microscope with a 100x Plan ApoChromat NA 1.4 oil lens. Images were acquired using the Photometrics Evolve EMCCD camera operated through Axiovision software. Images were processed and analysed using ImageJ software (National Institutes of Health).

## Live cell recombination assay

Yeast strains were placed on SPO agar plates (0.8% KAc and 2% agar) and allowed to sporulate at 300°C. After 4 days, images were captured in three channels and the pattern of fluorescence scored in the tetrads. To prevent confounding effects due to chromosome mis-segregation (a common occurance in kinetochore mutants), only tetrads where all 4 spores had acquired CFP (blue) fluorescence were included in the final analysis. Recombination frequency, expressed as map distance in Morgans, and standard error, was calculated using online tools (http://molbio.uoregon.edu/~fstahl/compare2.php). Power analysis was performed to determine sample size required for >0.87 confidence in differences from wild type and Fishers exact test was used to determine significance. All raw data and statistical analysis is given in *Supplementary files 1* and *2.*

## Southern blot analysis of DSB formation

Southern blotting was performed as previously described (*Vader et al., 2011*). The following probes (SGD coordinates) were used: *YCR047C;III*, 209,361–210,030. *CEN1:I*, 145,305–145,650. DSB intensities were analysed using ImageJ.

## Flow cytometry

Flow cytometry was performed as described (*Vader et al., 2011*).

## Genome-wide analysis of recombination and meiotic DSBs

Identification of single nucleotide polymorphisms by high-throughput sequencing was carried out as described by *Oke et al. (2014)*. Spo11-oligo maps of *mcm21Δ* were generated as described by *Zhu and Keeney, 2015* with modifications in sporulation culture cell density and Spo11-Flag immunoprecipitation (IP). Briefly, after 14 hr pre-sporulation in YPA media, cells were transferred to sporulation media (SPM) described in *Neale and Keeney (2009)* to a cell density ($OD_{600}$) of 6.0. The Spo11-oligo maps were generated from samples harvested after 4 hr in SPM. Spo11-Flag IP was carried out as described, except with protein G Dynabeads (Life Technologies, Carlsbad, California) instead of protein G agarose beads (400 μl protein G Dynabeads per 25 ml whole-cell extract in 50 ml IP volume for first round of IP; 125 μl protein G Dynabeads in 800 μl IP volume for second round of IP).

## Acknowledgements

This work was supported by the Wellcome Trust [090903], [092076] and [089396] (AM, NV and AK), a fellowship from the Netherlands Organisation for Scientific Research (NWO Veni-016.111.004) (GV), NIH/NIGMS grant R01 GM088248 (AH), NIH grant R01 GM097213 (JCF) and by the US National Institutes of Health grant R01-GM058673 (SK). The laboratory of GV is supported by a grant from the European Research Council (ERC StG 638197 "URDNA"). We are grateful to Bianka Baying at Genecore EMBL for library preparation and sequencing; Agnes Viale and the MSKCC Integrated Genomics Operation for sequencing Spo11 oligos; Nick Socci at the MSKCC Bioinformatics Core for mapping Spo11 oligos; Stewart Shuman (MSKCC) for gifts of T4 RNA ligase. We thank Julie Blyth for help with strain construction. GV acknowledges Gerry Fink for support during the initiation of this study. We thank Elizabeth Bayne, Puck Knipscheer, David Leach, Nuno Martins, Andrea Musacchio, Alex Bird, Arnaud Rondelet and John Weir for comments on the manuscript.

## Additional information

### Funding

| Funder | Grant reference number | Author |
| --- | --- | --- |
| Wellcome Trust | 092076 | Nadine Vincenten<br>Alastair RW Kerr<br>Adèle L Marston |
| Wellcome Trust | 089396 | Nadine Vincenten<br>Adèle L Marston |
| Wellcome Trust | 090903 | Nadine Vincenten<br>Adèle L Marston |
| National Institutes of Health | R01 GM088248 | Andreas Hochwagen |
| National Institutes of Health | R01 GM097213 | Jennifer Fung |
| National Institutes of Health | R01-GM058673 | Scott Keeney |
| Nederlandse Organisatie voor Wetenschappelijk Onderzoek | NWO Veni-016.111.004 | Gerben Vader |
| European Research Council | ERC StG 638197 URDNA | Gerben Vader |

The funders had no role in study design, data collection and interpretation, or the decision to submit the work for publication.

### Author contributions

NV, Designed and conducted live cell recombination, ChIP and microscopy experiments and generated strains for genome-wide mapping of recombination and DSBs, discussed results and commented on the paper, Conception and design, Acquisition of data, Analysis and interpretation of data, Drafting or revising the article; L-MK, Designed and performed all Southern blot DSB mapping experiments, Discussed results and commented on the paper, Conception and design, Acquisition of data, Analysis and interpretation of data, Drafting or revising the article; IL, Generated and analyzed genome-wide DSB maps; Discussed results and commented on the paper, Acquisition of data, Analysis and interpretation of data; AO, JF, Conducted and interpreted genome-wide recombination analysis, Discussed results and commented on the paper, Acquisition of data, Analysis and interpretation of data; ARWK, Analyzed ChIP-Seq data, Discussed results and commented on the paper, Acquisition of data, Analysis and interpretation of data; AH, Conceptually contributed to the study; Discussed results and commented on the paper, Conception and design, Contributed unpublished essential data or reagents; SK, Generated and analyzed genome-wide DSB maps, Discussed results and commented on the paper, Acquisition of data, Analysis and interpretation of data; GV, Designed and performed all Southern blot DSB mapping experiments; Designed the study, Analyzed the data and wrote the paper, Contributed equally to the study, Discussed results and commented on the paper, Conception and design, Acquisition of data, Analysis and interpretation of data, Drafting or revising the article; ALM, Designed the study, analyzed the data and wrote the paper, Contributed equally to the study, Discussed results and commented on the paper, Conception and design, Acquisition of data, Analysis and interpretation of data, Drafting or revising the article

### Author ORCIDs

Alastair RW Kerr, http://orcid.org/0000-0001-9207-6050
Scott Keeney, http://orcid.org/0000-0002-1283-6417

## Additional files

### Supplementary files

• Supplementary file 1. Raw data and analyses of map distances in the pericentromeric interval using the fluorescence live cell reporter assay.

• Supplementary file 2. Raw data and analyses of map distances in the arm interval using the fluorescence live cell reporter assay.

• Supplementary file 3. Analysis of genome-wide recombination. Source data used to generate *Figure 2* and *Figure 2—figure supplement 2*.

• Supplementary file 4. Yeast strains and qPCR primers used in this study. (A) List of yeast strains. (B) List of qPCR primers.

### Major datasets

The following datasets were generated:

| Author(s) | Year | Dataset title | Dataset URL | Database, license, and accessibility information |
|---|---|---|---|---|
| Nadine Vincenten, Lisa-Marie Kuhl, Isabel Lam, Ashwini Oke, Alastair RW Kerr, Andreas Hochwagen, Jennifer Fung, Scott Keeney, Gerben Vader, Adèle L Marston | 2015 | ChIP-Seq experiments to analyse Rec8 and Zip1 localisation in Ctf19c mutants during meiotic prophase | http://www.ncbi.nlm.nih.gov/geo/query/acc.cgi?acc=GSE70032 | Publicly available at the NCBI Gene Expression Omnibus (Accession no: GSE70032). |
| Nadine Vincenten, Lisa-Marie Kuhl, Isabel Lam, Ashwini Oke, Alastair RW Kerr, Andreas Hochwagen, Jennifer Fung, Scott Keeney, Gerben Vader, Adèle L Marston | 2015 | Data from: The kinetochore prevents centromere-proximal crossover recombination during meiosis | http://dx.doi.org/10.5061/dryad.22f52 | Available at Dryad Digital Repository under a CC0 Public Domain Dedication |
| Nadine Vincenten, Lisa-Marie Kuhl, Isabel Lam, Ashwini Oke, Alastair RW Kerr, Andreas Hochwagen, Jennifer Fung, Scott Keeney, Gerben Vader, Adèle L Marston | 2015 | Spo11-oligo mapping in S. cerevisiae Ctf19/CCAN kinetochore sub-complex mutant mcm21 | http://www.ncbi.nlm.nih.gov/geo/query/acc.cgi?acc=GSE72683 | Publicly available at the NCBI Gene Expression Omnibus (Accession no: GSE72683). |
| Nadine Vincenten, Baying B, Alastair RW Kerr | 2015 | ChIP-Seq experiment to analyse Zip1 localisation in the Ctf19c mutants during meiotic prophase | http://www.ncbi.nlm.nih.gov/geo/query/acc.cgi?acc=GSE70029 | Publicly available at the NCBI Gene Expression Omnibus (Accession no: GSE70029). |
| Nadine Vincenten, Baying B, Alastair RW Kerr | 2015 | Rec8 ChIP-Seq experiment to analyse cohesin localisation in the Ctf19c mutants during meiotic prophase | http://www.ncbi.nlm.nih.gov/geo/query/acc.cgi?acc=GSE70030 | Publicly available at the NCBI Gene Expression Omnibus (Accession no: GSE70030). |

The following previously published dataset was used:

| Author(s) | Year | Dataset title | Dataset URL | Database, license, and accessibility information |
|---|---|---|---|---|
| Ashwini Oke, Anderson CM, Yam P, Jennifer Fung | 2014 | Data from: Controlling meiotic recombinational repair: specifying the roles of ZMMs, Sgs1 and Mus81/Mms4 in crossover formation | http://dx.doi.org/10.5061/dryad.79hn1 | Available at Dryad Digital Repository under a CC0 Public Domain Dedication |

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
