## [Decision Letter]

Thank you for submitting your work entitled "The kinetochore prevents centromere-proximal crossover recombination during meiosis" for peer review at *eLife*. Your submission has been favorably evaluated by Jim Kadonaga (Senior Editor), Michael Lichten (Guest Reviewing Editor), and three reviewers.

The reviewers have discussed the reviews with one another and the Reviewing editor has drafted this decision to help you prepare a revised submission.

Summary:

Meiotic recombination is suppressed in the vicinity of the centromere in many organisms. Studies in budding yeast have shown that this suppression extends not only to crossovers, but to the double-strand breaks (DSBs) that initiate meiotic recombination. This study shows that the conserved Ctf19/CCAN complex, part of the kinetochore which is necessary for recruitment of pericentric cohesin, is required for this suppression, but suggests that it suppresses breaks and crossover through distinct mechanisms.

The consensus of the reviewers was that this paper brings new information to an important topic and, although the mechanism by which the Ctf19/CCAN complex suppresses DSBs and crossovers remains unclear, the findings of the paper are potentially of sufficient interest, but the following issues need to be addressed. In some cases, the additional work and/or analysis requested below could change some of the major conclusions of the paper, although it seems unlikely that the main conclusion will be affected.

Essential revisions:

1) Are other kinetochore complexes involved CO/DSB suppression? Further mutant studies may be beyond the scope, but the involvement of other kinetochore proteins should be considered and discussed. For example, is the MIND complex retained at meiotic centromeres in the absence of the Ctf19 complex?

2) A general concern involved the switching between different mutants and chromosomal regions in different experiments and figures. If at all possible, additional mutants should be included or additional analysis done so that there is more overlap between figures. For example, the addition of *ctf19-del* to the ChIP-qPCR in Figure 4 would provide a useful point of comparison to the anchor-away allele used in Figure 5, and thus would provide at least an indirect measure of the efficacy of the anchor away. Similarly, it would be useful to make a direct comparison between COs and DSBs in the same CEN8-associated interval in the *mms21-del* mutant background.

3) Inclusion of primary data and/or error bars to allow the reader to evaluate differences. For tetrad analyses, this means that the primary data should be included in a supplementary table, so that statistical power can be evaluated as well as significance of difference. Three examples: in Figure 1, pCLB2-IML3 is "n.s." relative to wild-type, but one suspects that this may be due to insufficient numbers of tetrads being analyzed, given the differences in frequencies shown. There is a similar concern with Figure 5. Finally, it appears that Ctf19 complexes are increasing crossovers in the arm interval as well, and given the differences (15 versus 25%) it should be easy to make these numbers statistically significant. Given that the spore color assay makes it possible to readily examine hundreds of tetrads, it is important to analyze enough to exclude biologically significant differences that might be masked due to small numbers. In addition, there exist online tools (http://molbio.uoregon.edu/~fstahl/) that allow calculation of standard errors for tetrad-derived map distances. Please be aware that, upon revision, if it still appears that insufficient tetrads have been analyzed, there will likely be a direct request to analyze more.

4) The quality of southern blots is below standard; if phosphorimaging was used, it is permissible to subtract general background if this will increase clarity. In critical cases, in particular the comparison between anchor-away at 0 and 3 hr, DSBs must be quantified. It is not at all clear that there is a quantitative difference between the impact of anchor-away 0 versus 3 hr on DSBs and on crossovers, but such a difference lies at the heart of the argument that Ctf19/CCAN depletion differentially affects breaks and crossovers.

5) The anchor-away experiments in Figure 5, which lie at the heart of the paper, are problematic in several ways:

A) The extent of Ctf19 depletion at kinetochores is not assessed-this is an important control to do.

B) It appears that Ctf19 anchor-away does not result in a loss of pericentric cohesin regardless of when it is executed. Similarly, it appears that there are significant losses of cohesin at CEN4 regardless of the time of execution. Therefore, if there really are differences in DSB repair under the two anchor-away conditions (see point 4, above), it is difficult to see how these can be attributed to differences in cohesin recruitment.

C) To recapitulate point 4, it is not at all clear that crossovers and DSBs are differently affected by the time when Ctf19 anchor-away is executed. Careful quantification is necessary to prove this point, and if the DSB analysis was only done once, additional replicates will need to be done.

6) While it is clear that the Ctf19 complex is involved in suppressing DSBs near centromeres, is there still residual suppressing in its absence? In addition to the "fold-change" graph in Figure 3, it would be useful to plot the mean Spo11-oligo signal as a function of distance from the centromere, comparing wild-type and *mcm21-del*. See Figure 4 in Buhler et al., 2007 for an illustration.

7) While it is clear that pCLB2-IML3 is a hypomorph, the authors need to address the fact that it displays a clear increase in centromere-adjacent crossovers in only 8 of the 16 yeast chromosomes. Are these data compromised by insufficient numbers? As with tetrad data, the actual number of events need to be reported, so that they can be vigorously analyzed for sufficient statistical power and statistical significance.

8) With regards to the experiments addressing Zip1 and the role of Ctf19 complex-loaded cohesin in recruiting Zip1:

A) Ctf19 complex mutants seem to disrupt not only initial Zip1 recruitment to centromeres, but subsequent synapsis. It's not clear why this should be the case, if arm recombination and cohesin loading are unaffected, and this should be addressed.

B) While *zip1-del* mutants increase pericentromeric crossovers (and possibly zip1-S75E, if sufficient numbers of tetrads are examined), their impact on pericentromeric DSBs is not addressed, and should be.

C) The *scc4-m35* mutant almost entirely eliminates Zip1 loading at centromeres, but increases pericentromeric crossovers to a substantially greater extent than does *zip1-del*, suggesting that pericentromeric cohesin might be doing other things to suppress crossovers other than recruiting Zip1. This should be considered and discussed, and an epistasis experiment should be considered if you truly believe that this is the main mode of pericentric crossover suppression.

---

## [Author Response]

*Essential revisions:*

*1) Are other kinetochore complexes involved CO/DSB suppression? Further mutant studies may be beyond the scope, but the involvement of other kinetochore proteins should be considered and discussed. For example, is the MIND complex retained at meiotic centromeres in the absence of the Ctf19 complex?*

During meiotic prophase, only the Ctf19/CCAN and MIND complexes are present at centromeres. It would be interesting to examine the role of the MIND complex in CO suppression, however, there are considerable technical difficulties with doing so and this is beyond the scope of the current study. MIND is known to be essential for kinetochore-microtubule attachments and, consistent with this, depletion of its Mtw1 and Dsn1 components during meiosis led to a near-complete failure in chromosome segregation with very few tetranucleate cells being produced (this data is now included as Figure 1—figure supplement 1). Since measurement of CO frequency relies on the production of spores, we were unable to assay CO suppression in these strains.We were, however, able to examine DSBs at *CEN1* by Southern blotting. We found that Dsn1 is also required for suppression of DSBs, suggesting the integrity of the kinetochore might be generally important for DSB suppression. This data is now included in Figure 4—figure supplement 1 and referred to in the last paragraph of the subsection “The Ctf19 complex inhibits centromere-proximal DNA breaks”.

The relationship between the Ctf19 complex and MIND is complex and currently unclear. We observed a modest reduction in the centromeric localization of the MIND component, Dsn1, in the Ctf19 complex mutant, *mcm21*△ . Conversely, Mcm21 was retained at centromeres in Dsn1 cells and appears elevated in Mtw1-depleted cells, despite massive chromosome mis-segregation in these cells (Figure 1—figure supplement 1). This suggests a partial and complex inter-dependence of the Ctf19 and MIND complexes at the kinetochore. The finding that Dsn1 is at least partially localized in *mcm21*△ cells is consistent with the fact that nuclear division occurs in *mcm21*△ but not *pCLB2-DSN1* cells and suggests that the Ctf19 complex functions at least partially independent of the MIND complex in CO suppression. Although we agree that it will be important to understand the relationship between the Ctf19 complex and MIND in CO suppression in the future, further analysis is beyond the scope of this study.

*2) A general concern involved the switching between different mutants and chromosomal regions in different experiments and figures. If at all possible, additional mutants should be included or additional analysis done so that there is more overlap between figures. For example, the addition of ctf19-del to the ChIP-qPCR in Figure 4 would provide a useful point of comparison to the anchor-away allele used in Figure 5, and thus would provide at least an indirect measure of the efficacy of the anchor away. Similarly, it would be useful to make a direct comparison between COs and DSBs in the same CEN8-associated interval in the mms21-del mutant background.*

We have now included *chl4*△ and *ctf19*△ strains in the analysis of Rec8 localization by ChIP-qPCR (Figure 5 in the revised version) to allow comparison with the anchor away experiments.

We now also include a direct comparison of DSBs and COs in the *CEN8* interval analysed in the live cell recombination assay. This is shown in Figure 3 and the relevant text is in the first paragraph of the subsection “The Ctf19 complex inhibits centromere-proximal DNA breaks”. This analysis shows that there is an increase in DSB levels within the *CEN8-*associated interval of about 5-fold over wild type. The total increase in CO formation within this interval was 21-fold over wild type (Figure 3 and Figure 1), lending further support for a dual effect of the Ctf19 complex on recombination outcomes at pericentromeric regions (see also further below).

*3) Inclusion of primary data and/or error bars to allow the reader to evaluate differences. For tetrad analyses, this means that the primary data should be included in a supplementary table, so that statistical power can be evaluated as well as significance of difference. Three examples: in Figure 1, pCLB2-IML3 is "n.s." relative to wild-type, but one suspects that this may be due to insufficient numbers of tetrads being analyzed, given the differences in frequencies shown. There is a similar concern with Figure 5. Finally, it appears that Ctf19 complexes are increasing crossovers in the arm interval as well, and given the differences (15 versus 25%) it should be easy to make these numbers statistically significant. Given that the spore color assay makes it possible to readily examine hundreds of tetrads, it is important to analyze enough to exclude biologically significant differences that might be masked due to small numbers. In addition, there exist online tools (http://molbio.uoregon.edu/~fstahl/) that allow calculation of standard errors for tetrad-derived map distances. Please be aware that, upon revision, if it still appears that insufficient tetrads have been analyzed, there will likely be a direct request to analyze more.*

We have included error bars and statistical analysis throughout.

We thank the reviewers for the helpful suggestions related to the tetrad data and have now done extensive further analyses, which have allowed us to strengthen our conclusions. As suggested, we have now used the online tool to calculate standard errors and these are given on all tetrad data throughout the manuscript. We have also determined statistical power for all tetrad data: all significant effects reported have a statistical power of at least 0.95, with the exception of the decrease in map distance within the arm interval in *zip1*△ cells (power of 0.90) and the modest increase in map distance within the pericentromeric interval when Ctf19 is anchored away after DNA replication (power of 0.87). Of note to the specific examples raised by the reviewers: (1) CO recombination in the pericentromeric interval is indeed significantly increased in *pCLB2-3HA-IML3* cells (Figure 1); (2) There is a significant, albeit modest, increase in pericentromeric COs when Ctf19-Frb is anchored away after DNA replication (Figure 6). (3) Further analysis of the chromosome arm interval in Ctf19 complex mutants did not provide evidence of a significant increase (Figure 1). All raw data is now included in [Supplementary-material SD1-data], together with power calculations and statistical analysis.

It is important to note that although in principle it is possible to score hundreds of tetrads in this assay, this is not feasible for many of the strains in practice. Firstly, expression of the spore autonomous markers relies on tetrad formation and the fraction of Ctf19 complex mutant cells that reach this stage is very low (between ~10-30% depending on the mutant). Furthermore, chromosome segregation is greatly impaired in Ctf19 complex mutants and to avoid confounding effects of mis-segregation, we can only score tetrads where each spore inherits one copy of the chromosome *VIII* (4 blue spores), a relatively small fraction of the overall spore count, which is already low.

Finally, we found no evidence for significant changes in CO frequency on the chromosomal arm region in Ctf19 complex mutants compared to wild type (Figure 1), though the values appear slightly elevated. Power calculations indicate that at least 8000 tetrads per individual strain (containing a faithfully segregated chromosome *VIII*) would need to be analyzed to determine significance for these differences. Due to the difficulties mentioned above, this would be a huge undertaking that would require weeks of analysis for a single strain.

*4) The quality of southern blots is below standard; if phosphorimaging was used, it is permissible to subtract general background if this will increase clarity. In critical cases, in particular the comparison between anchor-away at 0 and 3 hr, DSBs must be quantified. It is not at all clear that there is a quantitative difference between the impact of anchor-away 0 versus 3 hr on DSBs and on crossovers, but such a difference lies at the heart of the argument that Ctf19/CCAN depletion differentially affects breaks and crossovers.*

We have now repeated and replaced some of the southern blots to improve quality (Figure 4 and Figure 6), and for others we have subtracted general background to increase clarity. In the case of the timed anchor-away experiment (Figure 6), we have quantified DSB-intensities at *CEN1* at relevant time points and treatment regimes (quantification in Figure 6). This quantification shows that DSB intensities at *CEN1* upon anchoring-away Ctf19 after S-phase (t=0 hours) reach comparable levels as compared to DSB intensities (albeit of course only after a delay) that are measured upon anchoring-away Ctf19 before S-phase (t=0 hours). These similar levels of DSB formation soon after rapamycin addition but irrespective of the time in meiosis at which Ctf19 is anchored away, is in contrast with the significant differences in CO levels that we observed when anchoring-away Ctf19 using identical regimes (Figure 6). The shown southern analysis and quantification is a representative experiment; a similar experiment was performed that demonstrated comparable effects on DSB intensities and anchoring away of a different Ctf19 complex, Iml3 showed a similar effect on DSB formation. However, due to technical difficulties in quantifying these blots, in part due to weak specific signals together with high background signals, we have decided to only include the DSB quantification of the southern blot that is shown in Figure 6.

We would like point out that several additional lines of evidence provide strong support for two differential effects of Ctf19/CCAN on DSB formation and recombination, with only the latter depending on cohesin:

A) Our analysis of DSB levels (based on Spo11-oligo mapping) and CO levels within the CO interval close to *CEN8* (Figure 3) revealed a 20-fold increase in CO levels, whereas DSB levels only exhibit a 5-fold increase in *mcm21*△. This strongly indicates that Ctf19/CCAN influences CO repair in ways other than merely suppressing DSBs.

B) We found DSB formation at *CEN1* to be independent of Rec8 (Figure 6), and we also did not observe *CEN1*-associated DSB formation in *scc4-m35* expressing cells (Figure 6), despite the fact that we did find a significant effect on pericentromeric CO formation (Figure 5).

C) Finally, we note genome-wide maps of DSBs and of CO formation in wild type cells (Blitzblau et al., 2007; Buhler et al., 2007; Chen et al., 2008; Pan et al., 2011) have revealed a stronger suppression of COs as compared to DSB formation, again hinting at suppression of pericentromeric CO formation in addition to mere suppression of DSBs.

*5) The anchor-away experiments in Figure 5, which lie at the heart of the paper, are problematic in several ways:*

*A) The extent of Ctf19 depletion at kinetochores is not assessed-this is an important control to do.*

We have performed anti-GFP ChIP analysis on Ctf19-Frb-GFP and presented the results in the revised manuscript (Figure 6). This experiment shows efficient depletion of Ctf19 from the centromere when rapamycin is added either at 0h or at 3h.

*B) It appears that Ctf19 anchor-away does not result in a loss of pericentric cohesin regardless of when it is executed. Similarly, it appears that there are significant losses of cohesin at CEN4 regardless of the time of execution. Therefore, if there really are differences in DSB repair under the two anchor-away conditions (see point 4, above), it is difficult to see how these can be attributed to differences in cohesin recruitment.*

It is true that some centromeric cohesin is present in prophase even after anchoring away Ctf19 (Figure 6, this is also true in *ctf19*△ cells (Figure 5). Previous work in vegetatively growing cells (Fernius and Marston, 2009) demonstrated that the Ctf19 complex is important for the timely recruitment of cohesin to centromeres (i.e. before S phase), so that it can be converted to functional cohesin (i.e. cohesin that supports sister chromatid cohesion), which requires DNA replication, as previously demonstrated (Uhlmann and Nasmyth, 1998). In the absence of the Ctf19 complex, cohesin presumably moves from other (chromosome arm) loading sites to the centromere, but does not become functional because it arrives after the replication fork has already passed. Based on these observations, we hypothesized that only in the case where Ctf19 is present as cells undergo replication (i.e. rapamycin addition at t=3) would functional cohesin be built, while addition of rapamycin before DNA replication (t=0) would preclude functional cohesion establishment. During anaphase I, functional pericentromeric cohesin is protected from separase cleavage and can be observed at centromeres on chromosome spreads. As a measure of the functionalityof cohesin, we therefore performed chromosome spreads to visualize Rec8 in binucleate cells after release from a prophase I block after anchoring away Ctf19 at either t=0 or t=3. This data (now included in Figure 6) clearly shows an increase in pericentromeric cohesin in binucleate cells treated with rapamycin at t=3, compared to those treated at t=0. We have also extensively re-written this section more carefully explaining our rationale for clarity (subsection “Pericentromeric cohesin shields against CO repair of centromere-proximal DSBs”, second paragraph). Together with the quantification of DSBs (see point 4 above) and the finding that cohesin (*rec8*△ and *scc4-m35*) mutants do not relieve DSB suppression, we believe there is now strong evidence that the Ctf19 complex suppresses DSBs independently of cohesin.

*C) To recapitulate point 4, it is not at all clear that crossovers and DSBs are differently affected by the time when Ctf19 anchor-away is executed. Careful quantification is necessary to prove this point, and if the DSB analysis was only done once, additional replicates will need to be done.*

See point 4.

*6) While it is clear that the Ctf19 complex is involved in suppressing DSBs near centromeres, is there still residual suppressing in its absence? In addition to the "fold-change" graph in Figure 3, it would be useful to plot the mean Spo11-oligo signal as a function of distance from the centromere, comparing wild-type and mcm21-del. See Figure 4 in Buhler et al., 2007 for an illustration.*

This is a great suggestion. We have plotted the Spo11-oligo data as suggested, and this data is now presented as Figure 3. The graph clearly shows that there is no residual suppression of DSBs in the absence of the Ctf19 complex, effectively making the regions close to centromeres experience DSB levels that are comparable to average genome-wide DSB levels. We now also explicitly make this point in the text (subsection “The Ctf19 complex inhibits centromere-proximal DNA breaks”, first paragraph).

*7) While it is clear that pCLB2-IML3 is a hypomorph, the authors need to address the fact that it displays a clear increase in centromere-adjacent crossovers in only 8 of the 16 yeast chromosomes. Are these data compromised by insufficient numbers? As with tetrad data, the actual number of events need to be reported, so that they can be vigorously analyzed for sufficient statistical power and statistical significance.*

Due to poor spore viability of the *pCLB2-3HA-IML3* strain we were able to recover only 8 full tetrads for sequencing, so yes, the data are compromised by sufficient numbers and no statistical arguments can be made when considering the chromosomes individually. We included the data in Figure 2—figure supplement 2 for information only. We have now also included the source data we used to generate Figure 2 and Figure 2—figure supplement 2 as [Supplementary-material SD3-data]. All statistical arguments are made only when considering all chromosomes together and this analysis is also included in [Supplementary-material SD2-data].

*8) With regards to the experiments addressing Zip1 and the role of Ctf19 complex-loaded cohesin in recruiting Zip1:*

*A) Ctf19 complex mutants seem to disrupt not only initial Zip1 recruitment to centromeres, but subsequent synapsis. It's not clear why this should be the case, if arm recombination and cohesin loading are unaffected, and this should be addressed.*

We agree with the reviewers that our observation that full assembly of Zip1 along the spread chromosomes is impaired in Ctf19 complex mutants was unexpected and the reason remains unclear. We suggest that the lower percentage of cells with full length Zip1 staining along chromosomes is simply due to a reduction of Zip1 enrichment in the centromeric region, leading to the “discontinuous” appearance. Ctf19 complex mutants also exhibit a delay in meiotic G2/prophase which potentially exacerbates this effect. Furthermore, our Zip1 ChIP-Seq analysis revealed the strongest concentration of Zip1 at the centromere in wild type cells and a clear reduction in centromeric Zip1 in *mcm21*△ and *iml3*△ mutants. In contrast, chromosomal arm Zip1 does not appear to be grossly affected by deletion of *MCM21* or *IML3* in the ChIP-Seq or ChIP-qPCR analysis (with the exception of the *ctf19*△ mutant, for reasons that are unclear). Furthermore, previous work has established the centromere as the initiation site for chromosome synapsis (Tsubouchi et al., 2008 Genes Dev), yet the importance of this initiation site to the “full” Zip1 tracts typical of wild type cells has not, to our knowledge, been tested. Taken together, our Zip1 ChIP-Seq and chromosome spreading data indicate that initial Zip1 loading at the centromere may be important for robust Zip1 tracts along entire chromosomes. While indeed interesting, we feel that further investigation of the role of the kinetochore in synaptonemal assembly is beyond the scope of the current study but have added a couple of sentences highlighting this observation in the first paragraph of the subsection “Zip1 prevents pericentromeric COs and promotes centromere pairing through separate mechanisms”.

*B) While zip1-del mutants increase pericentromeric crossovers (and possibly zip1-S75E, if sufficient numbers of tetrads are examined), their impact on pericentromeric DSBs is not addressed, and should be.*

Blitzblau et al., 2007 and Chen et al., 2008 previously showed that Zip1 is not required to suppress pericentromeric DSBs globally, by mapping ssDNA, and also showed, using southern blotting, that DSB intensities and patterns did not change around individual centromeres. We have now used Southern blotting to analyze *CEN1-*proximal DSBs in *zip1*△ cells and come to similar conclusions: Zip1 is not essential for centromere-proximal DSB suppression. This data is included as Figure 7—figure supplement 1. This finding, together with the fact that we found no evidence that pericentromeric COs are increased in *zip1-S75E* led us to expect no increase in pericentromeric DSBs in the *zip1-S75E* mutant and therefore, given the limited time, we concluded that examination of DSBs in the *zip1-S75E* mutant was not an important priority.

*C) The scc4-m35 mutant almost entirely eliminates Zip1 loading at centromeres, but increases pericentromeric crossovers to a substantially greater extent than does zip1-del, suggesting that pericentromeric cohesin might be doing other things to suppress crossovers other than recruiting Zip1. This should be considered and discussed, and an epistasis experiment should be considered if you truly believe that this is the main mode of pericentric crossover suppression.*

We have now analyzed additional tetrads to achieve a statistical power of at least 0.96 and found only a minor difference in CO frequency within the pericentromeric interval between *scc4-m35* and *zip1*△ cells (2.4cM vs 2.0cM, respectively). However, this still does not rule out Zip1-independent functions for cohesin in CO suppression and we have included a statement to this effect (subsection “Zip1 prevents pericentromeric COs and promotes centromere pairing through separate mechanisms”, first paragraph).

We attempted to analyze a *scc4-m35 zip1*△ double mutant in the live cell recombination assay, but unfortunately these cells failed to undergo meiosis and produce tetrads. Although the reasons are currently unclear, this may suggest independent roles for cohesin and Zip1 in some aspect of meiosis.